



# The ISC-GEM Earthquake Catalogue (1904-2014): status after the Extension Project

Domenico Di Giacomo[1], E. Robert Engdahl[2], and Dmitry A. Storchak[1]

[1]International Seismological Centre (ISC), Pipers Lane, Thatcham, Berkshire, RG19 4NS, United Kingdom
[2]University of Colorado, Boulder, CO, USA

**Correspondence:** Domenico Di Giacomo (domenico@isc.ac.uk)

**Abstract.** We outline the work done to extend and improve the ISC-GEM Global Instrumental Earthquake Catalogue, which was first released in 2013 (Storchak et al., 2013, 2015). Due to time and resource limitations, version 1 (V1) of the ISC-GEM Catalogue included global earthquakes selected according to time dependent cut-off magnitudes between 1900 and 2009: 7.5 and above before 1918 (plus significant 6.5 and above continental earthquakes); 6.25 between 1918 and 1959; 5.5 from 1960 onwards. With the Extension Project we added both pre-1960 events below the original cut-off magnitudes (if enough station data was available to perform relocation and magnitude re-computation) and added magnitude 5.5 and above events from 2010 to 2014. The project ran over a 4-year period where a new version of the ISC-GEM Catalogue was released each year via the ISC website (www.isc.ac.uk/iscgem/). For each year, not only have we added new events to the catalogue for a given time range but also revised events already in V1 if additional data became available or location and/or magnitude reassessments were required. Here we recall the general background behind the production of the ISC-GEM Catalogue and describe the features of the different periods where the catalogue has been extended. Compared to the 2013 release, the new version (V5) of the ISC-GEM Catalogue now contains about 12,000 more events between 1904 and 1960 and ends in 2014 instead of 2009. We expect the ISC-GEM Catalogue to continue to be one of the most useful datasets for studies of the Earth's global seismicity and an important benchmark for seismic hazard analyses, and, ultimately, an asset for the seismological community as well as other geoscience fields, education and outreach activities. The ISC-GEM Catalogue is freely available at http://doi.org/10.31905/D808B825.

**Keywords:** earthquake location, magnitude, instrumental catalogue, parametric data collection, seismicity

## 1 Introduction

Earthquake catalogues are used in many activities by the seismological community. Usually these list basic focal parameters of seismic events (e.g., location, origin time, depth) along with the magnitude, and, eventually, other parameters (e.g., moment tensor or fault plane solutions). Studies concerning seismic hazard and the Earth's global seismicity often require as input an earthquake catalogue that (ideally) has been obtained using the same procedures over a long period of time. For such and other purposes, global instrumental earthquake catalogues have been produced by many authors since the beginning of the last century. Among others, catalogues from Gutenberg and Richter (1954), Båth and Duda (1979), Abe (1981, 1984) and Abe and



Noguchi (1983a, b), Pacheco and Sykes (1992), have been extensively used over the past decades until Engdahl and Villaseñor (2002) released the Centennial Catalogue, which now covers the period 1900-2007/09. Although such catalogues proved to be important resources for many years, they cover different time periods and, more importantly, are often characterized by either large heterogeneities in their parameters and/or produced with unknown procedures and/or underlying data (e.g., Di Giacomo et al., 2015a). For example, the Centennial Catalogue lists both locations from various catalogues (including the ones mentioned above) and re-computed ones using the Engdahl et al. (1998) methodology (normally referred to as EHB) for selected large earthquakes pre-1964 and from 1964 onwards, whereas magnitudes are not re-computed but compiled from several different sources/authors (see Di Giacomo et al., 2015a). In addition, most of those catalogues terminate at different times and are no longer maintained. In this context, in 2010 the International Seismological Centre (ISC, www.isc.ac.uk) as requested by the GEM Foundation (www.globalquakemodel.org), undertook a major effort to reprocess 100+ years of instrumental seismological data to reassess both locations and magnitudes of global (i.e., having magnitude 5.5 and above in our framework) earthquakes and, consequently, to produce a new earthquake catalogue using homogeneous and documented methodologies over the longest possible period of instrumental seismology (i.e., since the early 20$^{th}$ century). In January 2013, after a 27 months project, the ISC and a team of international experts (www.isc.ac.uk/iscgem/people.php) released on the ISC website (www.isc.ac.uk/iscgem/) the first version (V1, for a general description see Storchak et al., 2013, 2015) of the ISC-GEM Global Instrumental Earthquake Catalogue (1900-2009).

Since then the ISC-GEM Catalogue has been used by many researchers investigating seismicity rates, patterns of seismicity and earthquake forecast (e.g., Cambiotti et al., 2016; Geist, 2014; Ikuta et al., 2015; Kagan, 2017; Kagan and Jackson, 2016; Katsumata, 2015; Pollitz et al., 2014; Quinteros Cartaya et al., 2016; Roth et al., 2017; Zaliapin and Kreemer, 2017; Zechar et al., 2016; Zhan and Shearer, 2015) as well as by groups working on earthquake catalogues for seismic hazard purposes (e.g., Alvarez et al., 2016; Deif et al., 2017; Ghasemi et al., 2016; Kadirioğlu et al., 2016; Markušić et al., 2015; Mikhailova et al., 2015; Poggi et al., 2017; Weatherill et al., 2016) and other seismological studies (e.g., Lange et al., 2017; Leonard, 2014; Metzger et al., 2017; Ye et al., 2016).

In recognition of the value of such a homogeneous (to the largest extent possible) instrumental catalogue, funding from public and commercial organizations (www.isc.ac.uk/iscgem/sponsors.php) has been given to us since November 2013 to work on the extension of the ISC-GEM Catalogue over a 4-year project which, in a nutshell, aimed at adding as many earthquakes as possible before 1960 and continue the catalogue beyond 2009. The Extension project was also motivated by the fact that damaging pre-1960 earthquakes were below the cut-off magnitude of 6.25 (e.g., the 30 October 1930 Central Italy event, which caused collapse and severe damage in various towns) and many pre-1960 events had no initial magnitude and therefore could not be selected for V1, yet they could be large enough to be part of the ISC-GEM Catalogue.

Below we detail the work done during the 4-years of the Extension Project (ended in December 2017) and discuss features of the different time periods extended. Then we outline the overall state of the ISC-GEM Catalogue in its latest version (V5) and, finally, present the outlook for its further advancement.



## 2   The 4-year plan of the Extension Project

The Extension Project of the ISC-GEM Catalogue has been designed to add earthquakes smaller than magnitude 6.25 before 1960 and extend it beyond 2009 with events of magnitude 5.5 and above. In addition, many earthquakes pre-1960 having no magnitude information needed to be processed to reassess location and magnitude, if enough station data was available.

Fig. 1 summarizes the annual number of events before 1960 included in V1 of the ISC-GEM Catalogue along with the pre-1960 events available in the International Seismological Summary (ISS, 1918-1963, see also Villaseñor and Engdahl, 2005), BAAS (1913-1917) and the Centennial Catalogue plus additional hypocentres (hereafter we refer to it as augmented Centennial Catalogue) that were not processed for the V1 release (see also Figure 8 in Storchak et al., 2015). Note that ISS and BAAS earthquakes are also listed in the Centennial Catalogue but throughout the paper we try to refer to the original sources as much

as possible. For the sake of simplicity, in the following we refer to earthquakes in grey in Fig. 1 as extension events (i.e., not listed in V1), meaning that those are the events we digitize station data for but not necessarily all will be selected for processing and then included in the ISC-GEM Catalogue. The station data collection and selection issue will be discussed in the following sections.

The annual number of events in V1 oscillates between 4 and 12 for 1904-1917, 31 and 92 for 1918-1959, 235 and 489

for 1960-2009. Such variations reflect the cut-off magnitudes adopted for selecting earthquakes in different time periods: 7.5 and above before 1918 (plus significant 6.5 and above continental earthquakes); 6.25 between 1918 and 1959; 5.5 from 1960 onwards (see Di Giacomo et al., 2015b, for more details on the V1 earthquake selection criteria). It is worth remembering here that the cut-off magnitudes are simply thresholds set for selection purposes (not all pre-1960 events have known or reliable magnitudes) and should not be interpreted as completeness levels (variations of the completeness over different time periods

for V1 were briefly outlined by Di Giacomo et al., 2015a and investigated in more detail by Michael, 2014).

Considering the number of pre-1960 earthquakes available (nearly 21,000, i.e., about 2,000 more than the V1 release covering 1900-2009) in the ISS (1918-1959), BAAS (1913-1917) and augmented Centennial Catalogue for which we had to look for station data (and, consequentially, digitise), we planned to extend the catalogue following a 4-year schedule as shown in Table 1. Such a time frame was necessary to allow us to be as comprehensive as possible in the station data collection task

and also to assess the  60% of extension events that had no initial magnitude information (in our database), and, therefore, could not have been selected just using any cut-off magnitude criteria (details in the next section). Whereas, the extension of the catalogue beyond 2009 would benefit from the data concurrently released in the ISC Bulletin and would follow the original selection criteria (i.e., earthquakes with magnitude 5.5 and above).

At the end of each project year an upgraded version of the catalogue was made available for download at www.isc.ac.uk/

iscgem/. The catalogue is distributed in CSV format and is composed of two parts (the Main catalogue, also available as a KMZ file for use with Google Earth, and the Supplementary catalogue, the latter including events with either poor location and/or magnitude quality, see Storchak et al., 2015). Location parameters and magnitudes (either direct or proxy moment magnitude Mw, Di Giacomo et al., 2015a) come with formal uncertainties and quality flags (from A to D, denoting well and poorly constrained parameters, respectively), followed, if available, by the solution of the Global Centroid Moment Tensor (GCMT,



www.globalcmt.org, Dziewonski et al., 1981; Ekström et al., 2012). The criteria to assign the quality flags for location, depth and magnitude are summarized in Table 2. For the location quality flag we consider the secondary azimuthal gap (largest azimuthal gap filled by a single station, Bondár et al., 2004, hereafter referred to as SGAP), the eccentricity of the error ellipses (Bondár and Storchak, 2011) and the event location accuracy if it is of high confidence to become a candidate for the IASPEI

Reference List (GTCAND in Table 2, see Bondár and McLaughlin, 2009, and www.isc.ac.uk/gtevents). For the depth quality flag we consider the availability of very close stations (within 10 km, NSTA10) and in the local distance range (within 150 km, NSTAlocal), the depth constrained by depth-phases (if available, depdp in Table 2) and the location accuracy (GTCAND). For the magnitude quality flag we consider the author (GCMT or literature, Lee and Engdahl, 2015) for direct Mw values, whereas for Mw proxy based on our re-computed MS or mb (Di Giacomo et al., 2015a) the quality flag depends on combinations of the

magnitude value, type (MS or mb), uncertainty, number of stations used, and the uncertainty of Mw proxy.

One of the key features of the ISC-GEM Catalogue is that all events since 1904 have been reprocessed using instrumental station parametric data and the ak135 model (Kennett et al., 1995). To extend the catalogue, we followed the same steps and methodologies used to create V1, as described in:

- Di Giacomo et al. (2015b, and references therein) for digitizing from printed bulletins body-wave arrival times and ampli-
tudes/periods (of surface waves in particular) for the pre-1960 events to allow relocations and magnitude re-computation, respectively. For the extension events, the most important source of body-wave arrival times was the ISS, whereas amplitudes and periods were retrieved from individual station or network printed bulletins;

- Bondár et al. (2015) for the two-tier relocation approach, which benefits both from the EHB location algorithm (Engdahl et al., 1998) and the new ISC locator (Bondár and Storchak, 2011) used to constrain the depth and the epicentre, re-
spectively. As the EHB and ISC location algorithms are also used to cross-check each other, the location consistency is checked twice;

- Di Giacomo et al. (2015a) for the magnitude re-computation, particularly for the surface wave magnitude MS, which, in turn, is used as the basis for Mw conversion for most of the events pre-1960;

- Lee and Engdahl (2015) for the literature search of reliable and direct computations of seismic moment $M_0$ and, therefore,
of Mw, for events before the GCMT solutions start in 1976.

The data collection has been the most time-consuming task and indispensable ingredient not only to extend the catalogue but also to revise and better constrain solutions of events already in V1 (details in the next sections). Indeed, compared to the data collected for the V1 release, we made a significant improvement in the number of amplitude and period data digitized, particularly for MS re-computation, thanks both to additional bulletins donated (or lent) to the ISC from various institutions and

individuals (including the personal collection of N. Ambraseys, more details available at www.isc.ac.uk/iscgem/acknowledge. php) and station bulletins that were not processed for V1 due to time and resource limitations. Later we also show how the additional data gathered during the last four years helped us revise and better constrain the MS of pre-1960 earthquakes



already listed in V1. With the end of the Extension Project in December 2017, in the following we outline the improvements
and features of different time periods where the ISC-GEM Catalogue has been extended.

## 3 Year I to III of the Extension, 1920-1959

In this section we describe the work done in the first three years of the Extension project to add earthquakes in the historical
period between 1920 and 1959.

### 3.1 Station data collection and earthquake selection

The variations in the annual number of the extension events Fig. 1 are the result of various factors. For example, a significant
increase in the annual number of events can be seen in 1918 coinciding with the beginning of the ISS, whereas a dip in the late
1930s to mid-1940s is associated with the disruption caused by World War II (more details later) and another dip in the mid-
1950s is due to the censoring introduced by ISS procedures (more details at page 3 of the ISS, 1953) to reduce the workload.
The annual variations in the number of the extension events also introduces an issue in selecting earthquakes for the ISC-GEM
Catalogue. For example, between 1950 and 1952 there are between 782 and 1384 extension events in the ISS, and this is more
than the average number of earthquakes of magnitude 5.5 and above in the ISC-GEM Catalogue in recent years. This means
that a fraction of the extension events should not be part of the ISC-GEM Catalogue as they are below the cut-off magnitude of
5.5. However, since, as mentioned earlier, about 60% of such events have no magnitude information in our database, we could
not use the original magnitude criteria of 5.5. Thus, we decided to base our selection criteria using for each extension event
both the distribution of stations in the ISS and the number of stations contributing with amplitudes/periods for magnitude re-
computation. This required a major effort to digitize both all ISS pages (not available in any electronic format) and amplitude
and period pairs (of surface waves, in particular) from the station/network printed bulletins (Di Giacomo et al., 2015b) for all
extension events. Here we briefly summarize the station data collected for the extension events and highlight some features
that are relevant to the ISC-GEM Catalogue.

Fig. 2 shows the distribution of stations listed in the ISS for each decade (1920s, 1930s, 1940s, 1950s) color-coded by
their body-wave arrivals contribution to the extension events along with the annual number of stations and body-wave arrivals
digitized from the ISS. The number of stations listed in the ISS generally increased from the 1920s to the late 1930s before
World War II affected various seismic stations and it is only around 1953 that the station contribution improved significantly.
The box-and-whisker plot of Fig. 3 summarizes the median number of stations per event in each year. It shows that only a
limited number of stations (median number ranging from 9 to 26) are usually associated to the extension events until 1952,
whereas from 1953 onwards there is a general improvement in this respect (median number of stations ranging from 66 to 99).
Another relevant feature to point out is the uneven station distribution, with Europe showing the highest density particularly
before the 1950s, and the lack of stations in Africa and vast parts of the southern hemisphere.

Fig. 4 and Fig. 5, similarly to Fig. 2 and Fig. 3, show the distribution of stations contributing with amplitudes for each decade
and the median number of stations supplying amplitudes in each year, respectively. The number of stations reporting amplitudes





increased until World War II, dropped in the 1940s, and improved significantly from 1953. European and many Russian stations are the most important contributors to amplitude readings compared to stations in other continents, except for La Paz (LPZ, Observatorio San Calixto, Bolivia) and Riverview College (RIV, Sydney, Australia) from the Jesuit seismic network (Udías and Stauder, 1996). The number of stations per event contributing with amplitudes ranges from 0 to above 40, with the median

5    per year oscillating from 0 to 6 (Fig. 5).

We selected earthquakes considering combinations of the number of body-wave arrival times and the number of stations available supplying amplitude data. As our relocation approach (Bondár et al., 2015) relies largely on teleseismic observations, we first excluded events having no teleseismic phases (above 18° distance) and no or only one station contributing amplitudes, and then, depending on the period, events having less than two or three stations with amplitudes and a small number of body-

wave arrival times (less than 6 to 30, depending on time and distance). We ended up with such criteria after starting with a comprehensive list and incrementally removing earthquakes with station coverage and/or arrival times too poor to obtain a relocation. There are a few exceptions to such criteria that were applied on a case by case basis (e.g., deep events well recorded but with no amplitudes or events with two stations reporting amplitudes and few ISS phases available). Also, between 1953 and 1956 we processed all events in the ISS due to their small number. Out of the 19,341 extension events between 1920 and

1959 we relocated 11,572. The annual numbers are shown in Fig. 6, where the variations are linked to the state of the global network during those years and the operational practice changes at the ISS, as mentioned earlier.

## 3.2    Relocations

The relocation task was performed in a similar manner as described by Bondár et al. (2015). It had to be done to reassess previous hypocenters (ISS or other authors adopted by ISS) of the selected extension events. In Fig. 7 the box-and-whisker

plots of the defining stations (i.e., stations with at least one arrival time that constrains the location, hereafter referred to as NDEFSTA) and SGAP for each year are shown. The NDEFSTA gradually increases from the 1920s to the 1950s (except for the slight dip in the 1940s, for reasons explained earlier), whereas the SGAP gradually improves over time. This in general leads to improved confidence in locations. Fig. 8 shows the location and depth differences between the previous (ISS or authors adopted by ISS) and the ISC-GEM hypocentres. With a few large exceptions, median location differences range from about 100 km in

the 1920s to about 20 km in the late 1950s. With depth differences, one must consider that for 9,418 relocated extension events the original depth was unknown and nominally set to zero. Also, it is important to point out that about half of the relocated extension events have no depth phases, therefore for those the depth was assigned to a default depth resulting from the tectonic setting or nearby earthquakes. However, as already pointed out by Bondár et al. (2015), we remove the artefact of having most shallow earthquakes set at zero km depth.

We checked the reliability of the ISC-GEM relocations in terms of network coverage, deviation from the available hypocentres grouped for an event, performed a cross-check between the EHB and ISCloc algorithms and considered the nearby seismicity. At times we also used available comments in the individual station bulletins as a guide in solving uncertain cases. Obviously, relocations for events with large SGAP (> 270º) and/or small NDEFSTA are not well constrained and we decided case by case whether to manually assign location flag D (i.e., the event will be listed in the Supplementary Catalogue). A




typical case in this respect (although time-dependent) is represented by earthquakes in the North Atlantic ridge where most of the phase data would come from European stations and SGAP could be even larger than 300º simply because North American stations (see Fig. 2) would not systematically report data for such earthquakes (except for large ones).

Table 3 summarizes the location and depth quality flags for the relocated extension events between 1920 and 1959. The most recurrent quality flag both for location and depth is C. However, despite the limitations of the global seismic network, particularly before the 1950s, it is possible to recognize the improvements of the ISC-GEM locations with respect to the original ones even on a global scale, as shown in Fig. 9. Although we do not claim that the ISC-GEM locations are the best possible solutions in this period for every single event, we recommend that any regional or focused study of historical earthquakes instrumentally recorded should start from the ISC-GEM locations as they are obtained from (currently) the most comprehensive set of instrumental data.

### 3.3 Magnitude re-assessment

We used the approach described in Di Giacomo et al. (2015a) to reassess the magnitude of the extension events consistently with their ISC-GEM relocations. Due to the lack of short period body-wave amplitudes before the 1960s, here we focus on re-computed MS as the basis for the calculation of the proxy Mw. The MS re-computation is based on the amplitudes and periods of surface waves digitized during this work (Fig. 4 and Fig. 5). Before accepting an MS value, we checked the station distribution and, when possible, cross checked our magnitudes with other magnitude information to investigate cases of large differences with previous results. Fig. 10 shows the time line of the re-computed MS and their annual counts. Besides the recurrent features discussed earlier (i.e., general increase in the annual counts from the early 1920s and the dip in the 1940s), there are 2,304 events with MS below 5.5. This occurs because our selection criteria for this period, as explained earlier, had to be based on station data availability rather on magnitude. Although events with magnitude below 5.5 would not normally be part of the ISC-GEM Catalogue, we did not exclude them because of the importance of re-assessing the magnitude of historical earthquakes. Most of these events with MS < 5.5 are mostly located in an area covering the mid-oceanic ridge of the North Atlantic to the Euro-Mediterranean region. This is not surprising considering the distribution of stations contributing amplitudes (Fig. 4). Also, there are 80 earthquakes with MS ≥ 6.5 that should have already been in V1. These events were not originally selected because the available magnitude information was considered not reliable or it was below the cut-off value of 6.25. This further highlights the necessity of a comprehensive and systematic magnitude re-assessment with homogeneous procedures.

In total, we re-computed MS for 6,575 ( 57%) of the relocated extension events and obtained a magnitude (MS or any other type) for the first time (at least to our knowledge) for 3,011 of them. A lack of stations reporting amplitudes is normally the cause for not having a re-computed MS as we normally require a minimum of three stations. The only exception occurs when we have two station magnitudes from a subset of specially selected stations that do not differ more than 0.3 magnitude units (m.u.). In such circumstances we allowed MS re-computation for 276 earthquakes and assigned MS uncertainty of 0.5 m.u.

If no direct Mw value is available for an event, the re-computed MS values are then used as the basis for proxy calculations of Mw and magnitude quality flags (Di Giacomo et al., 2015a). Table 4 summarizes the counts for the magnitude quality flags

for the relocated extension events between 1920 and 1959. Only five of the extension events have direct values of Mw from the literature search (Lee and Engdahl, 2015). The high number of magnitude quality flags D is largely due to events where no re-computed magnitude (MS, mb or Mw from the literature) is available and where MS, as the basis for Mw conversion, is below 5. Fig. 11 shows the time line of the earthquakes without re-computed magnitudes along with their annual counts and depth frequency. Although MS is not estimated for deep earthquakes according to IASPEI (2013), the clear majority (nearly 70%) of events without magnitude are shallow (depth $\leq$ 50 km). For such shallow earthquakes we continue to look for additional amplitudes (more details in a later section) so that we can calculate MS and eventually move some of those events from the Supplementary to the Main catalogue.

## 4    Year IV of the Extension, 1904-1919

During the last year of the Extension project we focused on the first part of 20[th] century and made special efforts to gather not only body-wave arrival times and amplitude of surface waves, but also known earthquakes not available in the ISC database. We did not add any station data before 1904 (basically only stations belonging to the Milne network are available, see, e.g., Adams, 1989) and, consequently, we decided to drop from the ISC-GEM Catalogue the ten pre-1904 events listed before V5 and have the catalogue starting in 1904.

### 4.1    Data collection

Before the ISS was put in production starting with earthquakes that occurred in 1918, other seismic bulletins were compiled by different authors/agencies (e.g., Schweitzer and Lee, 2002; Storchak et al., 2015, and references therein). For this work we gathered station data from:

- International Seismological Associations (ISA, 1904-1908) bulletins. These bulletins are the most comprehensive both in terms of earthquakes and stations listed for those years. They are composed of two parts, one for the large/significant earthquakes (in German "Hauptbeben") and one for the small ones ("Kleinere Beben"). Unfortunately, the 1908 "Hauptbeben" part was not printed (at least to our knowledge). Referred to as ISA in the following;

- Shide Circulars (BAASSC, 1900-1912) for 1908-1912, referred to as SHIDE;

- Gutenberg's notepads for 1908-1916 (mostly up to 1912), referred to as GUTE;

- Russian network bulletins for 1908 and 1911-1912, referred to as RUS;

- BAAS (1913-1917) bulletins (predecessor of the ISS) listing both locations and station data;

- ISS in 1918-1919;

- Individual station bulletins (1904-1919, referred to as BULLETINS) this time not only for what concerns the surface wave amplitudes but also for body-wave arrival times (in support of relocation rather than only magnitude re-assessment) as





some stations (e.g., Uppsala, Nordlingen, Munich) were partially or completely missing in the BAAS or ISS. Furthermore, the body-wave arrival times from individual station bulletins are fundamental for the newly added earthquakes that we describe later in this section.

The ISA, SHIDE and RUS bulletins are available from the supplementary material of Schweitzer and Lee (2002), whereas scanned images of GUTE notepads were kindly provided by K. Abe. The ISA, BAAS and ISS bulletins list arrival times from most of the stations operating at that time, whereas SHIDE includes mostly data from Milne stations and the GUTE notepads only a subset of global stations. Except for ISS 1918-1919 (already electronically available), the various sources of body-wave arrival times (ISA, SHIDE, GUTE, RUS and BAAS) for the 1904-1917 extension events were all manually typed in text files and then parsed into the ISC database.

As shown in Fig. 1, the annual number of recorded earthquakes, at least up to 1917, is smaller than an approximate average rate of ~100/year for events of magnitude 6 and above. Therefore, for this period we also tried to add as many known earthquakes as possible that are not listed in the augmented Centennial Catalogue, BAAS or even the ISS 1918-1919. To do that we considered the following sources:

- Catalog of Damaging Earthquakes in the World (http://iisee.kenken.go.jp/utsu/index_eng.html, Utsu, 1990, 2002, 2004), referred to as UTSU in the following;

- ISA (only for 1904-1907);

- SHARE European Earthquake Catalogue (SHEEC) 1900-2006 (Grünthal et al., 2013), referred to as SHEEC in the following;

- Karnik (1971) catalogue of the European area (referred to as KAR) and Papazachos et al. (2000, 2010) catalogue for Greece and surrounding areas (available at http://geophysics.geo.auth.gr/ss/CATALOGS/seiscat.dat, referred to as GRE) for earthquakes before 1908 with station data in ISA (either not available in SHARE or where the KAR/GRE solution would be a better starting point considering the ISA station data);

- Significant Earthquake Database of the National Geophysical Data Center / World Data Service (NGDC/WDS) (https://www.ngdc.noaa.gov/nndc/struts/form?t=101650&s=1&d=1), referred to as NGDC.

As we have a rather mixed set of starting points for hypocenter relocations, in Fig. 12 we show the time lines of the extension earthquakes 1904-1919 split by original location author along with their counts and the time coverage of the station data sources we digitized. The augmented Centennial Catalogue location sources G&R, B&D, ABE, CENT and BJI are from Gutenberg and Richter (1954), Båth and Duda (1979), Abe (1981, 1984) and Abe and Noguchi (1983a, b), Centennial itself and Chinese catalogue, respectively. In total we have found 405 additional earthquakes (mostly before 1917) on top of the 1,530 earthquakes already listed between 1904 and 1919 in the augmented Centennial Catalogue, BAAS and ISS. Notably, between 1904 and 1907 the annual number of earthquakes we added (mostly from ISA and UTSU) is larger than the annual number of extension earthquakes previously available in our record. Between 1908 and 1912 the annual number of earthquakes added is comparable



or smaller than the ones already available, whereas from the beginning of the BAAS and then ISS the annual number of newly added earthquakes drops significantly during the BAAS and then it is zero with the beginning of the ISS.

For all earthquakes outlined in Fig. 12 we tried to associate as many body-wave arrival times and surface wave amplitudes as possible from the station data sources mentioned earlier. The contribution of each station data source is presented in Fig. 13. For the early years of the past century, ISA was comprehensive in compiling data from stations around the world, whereas the other sources included only subsets of the stations operating at that time. Unfortunately, between 1908 and 1912 (coinciding with the end of ISA, "Hauptbeben" part, in 1907 and before the beginning of BAAS in 1913) we do not have a comprehensive bulletin such as ISA in preceding years or BAAS in the following ones. Therefore, we gathered station data from SHIDE, GUTE RUS and individual/network station bulletins. From 1913 onwards, the overall station data collection improves significantly thanks to BAAS and then ISS.

Fig. 14 shows, considering all sources depicted in Fig. 13, the overall annual counts of number of stations, phases and the box-and-whisker plot of the number of stations per event in each year. A significant dip is present in the station data between 1908 and 1912 since the station (and location) sources available to us for these years are not as comprehensive as ISA or BAAS/ISS. The box-and-whisker plot of Fig. 14 also shows that several earthquakes have none to three stations associated (59 from the augmented Centennial Catalogue, BAAS and ISS and 116 from the newly added ones). Obviously, the limitations in the collection of station data influenced the earthquakes that we finally selected for processing and the quality of the relocations/magnitude re-assessment. The results are discussed in the next two subsections.

## 4.2 Relocations

Not all extension earthquakes have sufficient station data to perform a relocation using our approach. First, we have discarded 175 earthquakes having less than four stations, as pointed out earlier. We then progressively discarded another 650 as either the relocation failed or was considered unreliable. We may to go back to the discarded earthquakes if additional station data becomes available to us. In the end, we accepted the relocation for 1,110 out of the 1,935 extension earthquakes. Fig. 15 shows the annual counts of the relocated extension earthquakes 1904-1919. Note the dip in the annual number of the relocated extension earthquakes for 1908-1912, reflecting the absence (to our knowledge) of a comprehensive global bulletin between ISA and BAAS.

As in Fig. 7, Fig. 16 shows the box-and-whisker plots of NDEFSTA and SGAP. For this period the relocations are usually based on a small number of stations (median between 6 and 16) resulting in a large SGAP (median between 201° and 310º), even during the years covered by BAAS and ISS. Fig. 17 shows the median location, depth and origin time differences between previous (see Fig. 12) and ISC-GEM locations. The median location differences oscillate between 70 and 205 km, with large differences above one thousand km for 46 earthquakes (16 above 2000 and 4 above 3000 km). Such large location differences can occur for various reasons (from typos in the latitude/longitude of previous locations to poorly recorded earthquakes having low confidence locations). One extreme example is the epicentre change from Bristol Bay, off-shore Alaska, (G&R location) to off-shore Jamaica (ISC-GEM location) for an event that occurred the $22^{nd}$ August 1907 (~$22^h23^m$). The reason for such a large difference originates from the fact that G&R ignored the report that the event was felt in Kingston (see, e.g., ISA, 1907,



part B, p. 73) and preferred to fit the phase data to an intermediate-depth event off-shore Alaska. As for 1920-1959, most of the earthquakes have no depth resolution and the previous depths were largely unknown or set to zero and this occasionally results in large depth changes (±100 and ±300 km for 51 and 10 earthquakes, respectively). Fig. 17 also shows the box-and-whisker plot of the origin time (OT) differences in each year. We show the OT differences because in this period (particularly before

BAAS) the OT listed in the previous location sources was at time truncated to the minute or with some minute error that we were able to address thanks to the stations data we digitized. Although 90% of the OT differences are within 1 minute, some large OT changes of ±5 minutes or more occur for 16 earthquakes (eight originally from ABE).

Similar to the 1920-1959 period, we assigned location quality flag D if the location was not constrained well enough. This time this task was done not only by considering the usual criteria (see 3.2) but also consulting available information on the
earthquake's effects (e.g., tsunami, damage). In this respect we made systematic use of the earthquake effect information available in UTSU and NGCDC. Table 5 summarizes the location and depth quality flags for the relocated extension events between 1904 and 1919. The limitations of the global network in this period are generally more prominent than for 1920-1959 and this translates in most of the earthquakes having location and depth quality C and about 246 of them have location quality D. As for the discarded earthquakes, if additional station data becomes available we will try to improve the location quality
and eventually move some of the location flag D earthquakes from the Supplementary to the Main catalogue. As for Fig. 9, Fig. 18 compares the previous (before) and ISC-GEM locations (after) on global maps where, again, a general improvement in the earthquakes distribution along plate boundaries is delineated. This is particularly the case for several global earthquakes along the subduction zone of the Pacific and Indian oceans whose previous locations were hundreds of km away from plate boundaries.

**4.3   Magnitude re-assessment**

Even for this period the magnitude re-assessment is mostly based on our re-computed MS. Following the same procedures described earlier, we obtained 927 MS for the relocated extension earthquakes, as shown in Fig. 19. For 500 of them we have computed a magnitude for the first time (in our record). Notably, for 137 earthquakes MS < 5.5, whereas MS ≥ 6.5 for 306 of them and > 7.5 for 12 of them. The latter includes six earthquakes originally from GUTE, four from ABE and two from
BAAS that were not selected for V1 because the magnitudes available were not considered reliable or were below 7.5 (the original cut-off magnitude for the V1 selection before ISS started in 1918). Nearly all earthquakes with MS < 5.5 occurred in the Euro-Mediterranean area (because in this period the stations contributing with surface wave amplitudes are strongly concentrated in Europe, see Fig. 13). For 1904 we were able to re-compute MS for three earthquakes as before December 1904 we could gather amplitudes only from stations GTT (Göttingen) and POT (Potsdam). For 18 earthquakes between 1905
and 1919 we accepted MS based on two station magnitudes. Except for four earthquakes where we have direct Mw values from the Lee and Engdahl (2015) literature search, all re-computed MS values are used as the basis for Mw proxy calculations (Di Giacomo et al., 2015a). Table 6 summarizes the counts for the magnitude quality flags for the relocated extension events between 1904 and 1919. About 50% of the 183 relocated extension earthquakes where we do not have a magnitude (no direct Mw or re-computed MS) are deep (MS not allowed in our procedures).





## 5 Summary of the Extension for 2010-2014

The extension of the ISC-GEM Catalogue beyond 2009 (last year in V1) benefits from the data already available in the ISC Bulletin and the review of global earthquakes by ISC analysts. The earthquake selection for recent years is based on magnitude (5.5 and above). Table 7 summarizes the number of earthquakes added per year during 2010-2014. The relatively high number of earthquakes in 2011 is due to the 11th March Mw = 9.1 Tohoku earthquake that was followed by about 120 aftershocks with magnitude 5.5 and above just in the first 24 hours. In contrast to the historical period, recent years global earthquakes are recorded by a dense global network that usually allows us to constrain the location with hundreds of stations and a relatively small SGAP. This is shown in Fig. 20 (note the difference in scale for the number of stations plot compared to Figs 7 and 16). The ISC-GEM epicentres do not move significantly from the previous ones (ISC locations) although occasional significant changes in depth occur, as shown in Fig. 21.

As to magnitude, we largely list direct Mw from GCMT (2,347 earthquakes). Proxy Mw from re-computed MS or mb are given for 248 earthquakes. The location and magnitudes of these earthquakes will be included in the figures of the section outlining the state of V5.

## 6 Review of events that have already been part of the catalogue

The ISC-GEM Catalogue comes with a version number that keeps track of the catalogue's updates and/or additions. Even when an earthquake is listed in the catalogue, we continue to look for additional station data and information that could help us to improve, whenever necessary, the earthquakes parameters we list in the catalogue. At the same time, we cooperate with users of the catalogue who inquire about earthquakes of their interest in different parts of the world, at times resulting in an updated location, depth and/or magnitude for one or more earthquake. Examples of updates we made thanks to user's help are available at the ISC-GEM Catalogue update log webpage (www.isc.ac.uk/iscgem/update_log). We also run internal checks as progress is made with the Rebuild of the ISC Bulletin (Storchak et al., 2017) and/or the ISC-EHB dataset (Weston et al., 2018). We try to keep the number of releases to a minimum and recommend users quote the version number when using the ISC-GEM Catalogue for their studies.

As mentioned before, during the Extension Project we gathered station data (particularly for amplitudes of surface waves) from printed station bulletins that were not available to us. Therefore, during the data collection task of the Extension Project we did not limit the search for amplitude data to extension earthquakes but also to earthquakes that were already listed in previous version (before V5) of the catalogue. This way we revised the MS of earthquakes already listed in the catalogue even if we added just one or two station readings. Fig. 22 shows the number of stations contributing to MS as well as the comparison between original and revised MS for pre-1960 earthquakes already listed in previous versions of the catalogue. The increase in the number of stations contributing to the re-computation of MS is significant: 30% and 74% of the original MS were constrained using less than six and eleven stations, respectively, whereas with the revised MS these percentages drop to ~8.5% and 31%. Also, the station data added allowed us to gain about 50 earthquakes with MS. About 97% of the revised MS are





within ±0.3 m.u. of the original ones with only five earthquakes having MS differences above ±0.6 m.u, (often due to originally mis-associated readings, also resulting in the loss of four original MS values).

## 7  Overall status of the new (V5) ISC-GEM Catalogue

The new version (V5) of the ISC-GEM Catalogue covers the period 1904-2014 and was released on the ISC website (www.
isc.ac.uk/iscgem/) 27$^{th}$ February 2018. It is composed of 35,225 earthquakes in total, with 7,126 listed in the Supplementary catalogue (about 93% of them having occurred before 1960). The magnitude sources are the same four described in Di Giacomo et al. (2015a) and the updated composition is as follows: 45.72% for Mw from GCMT, 42.85% for Mw proxy from re-computed MS, 8.1% for Mw proxy from re-computed mb, and, finally, 3.33% for Mw from the literature search. As outlined in Di Giacomo et al. (2015a), the Mw proxy values based on MS are mostly for pre-GCMT earthquakes (i.e., with few exceptions, before 1976).

The primary use of the ISC-GEM Catalogue is seismic hazard (including calibration of regional seismic catalogues) and Earth's seismicity pattern studies as is it the longest and most homogeneous record of natural global seismicity recorded during the instrumental period. For this reason, in Fig. 23 we plot V5 of the ISC-GEM Catalogue using Agnew (2014) symbols to emphasize the magnitude of the earthquakes in the catalogue. To produce the figure, we included earthquakes in the Main catalogue plus those earthquakes in the Supplementary catalogue that have reliable magnitude but poor location (i.e., magnitude quality flag not equal to D and location/depth quality flag equal to D). The subduction zones and the mid-oceanic ridges are well depicted as are areas where global earthquakes occur more frequently.

The current magnitude content as well as a basic magnitude completeness (Mc) assessment is shown in Fig. 24 (update on Figure 20 of Di Giacomo et al., 2015a). It is not our aim to do a detailed completeness study as Michael (2014), here we use the magnitude content and Mc to highlight the following features of the catalogue:

- the historical period (before the beginning of the ISC Bulletin in 1964) is not as complete (average annual Mc varying between 5.7 and 6.8) as more recent decades (average annual Mc between 5.5 and 5.7 since 1964). Important fluctuations in the annual number of earthquakes / Mc are present in specific periods or years. For example, because of World War II there is a significant decrease in the number of recorded earthquakes (particularly below magnitude 6) consistent with the disruption of the global network during the 1940s; other minor fluctuations are present in almost every decade (e.g., slight rise in Mc in the early 1960s and late 1970s);

- the number of intermediate-depth (between 60 and 300 km) and deep ($\geq$ 300 km) earthquakes per year before the 1950s-1960s is significantly smaller compared to more recent decades. The reason is not fully clear and will be a matter for further investigation (see 8). Most likely, it is the result of a combination of factors, which include the detection capability for moderate deep-focus earthquakes of analog seismographs (see, e.g., Kanamori, 1988) deployed around the world before the 1950s, the lack of stations close to subduction zones for many decades (Figs. 2 and 13), and the earthquake selection criteria. For global earthquakes, instruments such as the Wiechert, Bosh-Omori, Maika and Galitzin



were able to record surface wave signals (medium period range, centred around 20 s) better than body-waves (higher frequency signals, particularly P-waves, from around 10 s and below). The effect could have been that many stations would not report station data for moderate deep-focus earthquakes and, therefore, the ISS would not compile data for such earthquakes (i.e., the earthquake would not be recorded). The selection criteria could also play a role although the earthquakes not selected for processing either lack station data (and depth resolution) or, more importantly, are usually too small to account for the small number of deep-focus earthquakes depicted in Fig. 24.

## 8  Outlook

We plan to continue maintaining the ISC-GEM Catalogue for years to come and work on its advancement by:

- adding recent years (2015 onwards);

- regularising the magnitude for earthquakes between 1960 and 1990 to remove as many fluctuations as possible in the Mc over those decades;

- adding earthquakes between magnitude 5 and 5.5 that have occurred in continental areas from 1960 onwards;

- improving the content for the historical period (before 1964) by filling gaps in the station reports (particularly for what concerns surface wave amplitudes) and possibly bringing additional earthquakes and station data from the Bureau Central International de Séismologie (BCIS, 1933-1968); we will also consider any other source (if available) not considered so far that will bring useful data (station data and/or earthquake information) that will allow us to improve the catalogue; with this task we aim at moving as many earthquakes as possible from the Supplementary to the Main catalogue (see Fig. 25;

- integrating the results from the ISC Bulletin Rebuild project (1964-2010, see Storchak et al., 2017) and the ISC-EHB reconstruction (1964 onwards, Weston et al., 2018);

- continuing and extending our literature search for new or updates of direct estimation of Mw for pre-GCMT earthquakes as well as general focal parameters; we also aim at including fault plane solutions from the literature for historical earthquakes.

A more detailed description of the Advancement Project of the ISC-GEM Catalogue is available at www.isc.ac.uk/iscgem/advancement.php. We will continue releasing a new version after the end of each year of the Advancement Project. In this way we will be able to provide the seismological, as well as a broader geoscience community, with the most comprehensive and homogeneous account of earthquake global seismicity recorded instrumentally at any point in time.





## 9   Data availability

Since 27<sup>th</sup> February 2018, V5 of the ISC-GEM Catalogue has been available for download at http://doi.org/10.31905/D808B825. All data used in this paper are maintained at the ISC (www.isc.ac.uk). The ISC-GEM Catalogue is released without the associated seismic wave arrival times and amplitudes used for this work. These underlying parametric data are either already

available or will be before the end of 2018 as part of corresponding events in the ISC Bulletin (www.isc.ac.uk/iscbulletin/).

## 10   Conclusions

We presented the procedures and results of a 4-year project which extended and improved the ISC-GEM Catalogue first released in 2013 (Storchak et al., 2013). We have added about 12,000 more events between 1904 and 1960 and the new version (V5) ends in 2014 instead of 2009. To extend the catalogue before the 1960s we have digitized 650,000 phase arrival times

from various sources (ISS, BAAS, ISA, Shide circulars, Gutenberg notepads, etc.) in different periods and added 140,000 amplitudes from printed station bulletins. The features and limitations of the global network before 1960 have been outlined and the results show that the relocations, based on our two tier approach (Bondár et al., 2015), provide solutions distributed along main tectonic boundaries, even though they are usually based on a small number of stations compared relocations of earthquakes in recent years. We have re-computed over 6,000 MS values for pre-1960 earthquakes and obtained (to our record)

a magnitude for the fist time for more than 3,000 of them. For the period 2010-2014 we have greatly benefited from both the station data available in the ISC Bulletin and the reviews done by ISC analysts which provide us with robust starting points for the relocations and the Mw from the GCMT.

At the same time as the digitization from printed sources of stations supplying amplitude data (of surwace waves in particular), we also looked for additional data for historical earthquakes (pre-1960) already listed in previous versions of the

ISC-GEM Catalogue. The newly added amplitude data made us revise a significant number of pre-1960 earthquakes listed in V1 and improve the magnitude solutions as the revised magnitudes are now based on a much higher number of stations.

The current state of V5 of the ISC-GEM Catalogue has been summarized and its features outlined. With the Advancement project we aim to further improve and extend the catalogue in coming years and address some of the limitations that have been pointed out here during different periods of time.

*Competing interests.* The authors declare no competing interests in the production of the ISC-GEM Catalogue.

*Acknowledgements.* This project was supported by the NSF (Award 1417970), USGS (Award G15AC00202), FM Global, OYO Corporation, the Lighthill Risk Network, the Aspen Re, Bundesanstalt für Geowissenschaften und Rohstoffe (BGR), and 65 members of the ISC (www. isc.ac.uk/members/). Year I and II of the Extension project were also supported by the GEM Foundation. Daniela Olaru and Elizabeth Ayres were instrumental in the data collection from printed station bulletins and the ISS. We used computer codes by Antonio Villaseñor to digitize



the ISS phase data. We are deeply indebted to various institutions and individuals that provided additional station bulletins to the ISC (more details at www.isc.ac.uk/iscgem/acknowledge.php). We are grateful to Joseph Battló for sharing additional data on the 1919 Torremendo (Spain) series (Batlló et al., 2015) and related discussions which allowed us to correct the corresponding events originally listed in the ISS. For the Mw literature search we thank Paolo Harabaglia for pointing out the papers from Pino et al. (2000, 2008) for two significant

5   earthquakes in Italy (12 December 1908 Messina and 23 July 1930 Irpinia earthquakes) as well as the centroid solution for the 29 November 1975 Hilo (Hawaii) earthquake from Nettles and Ekström (2004). Further acknowledgments to users of the ISC-GEM Catalogue are available at www.isc.ac.uk/iscgem/update_log/. Figures were drawn using the Generic Mapping Tools (Wessel et al., 2013).



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





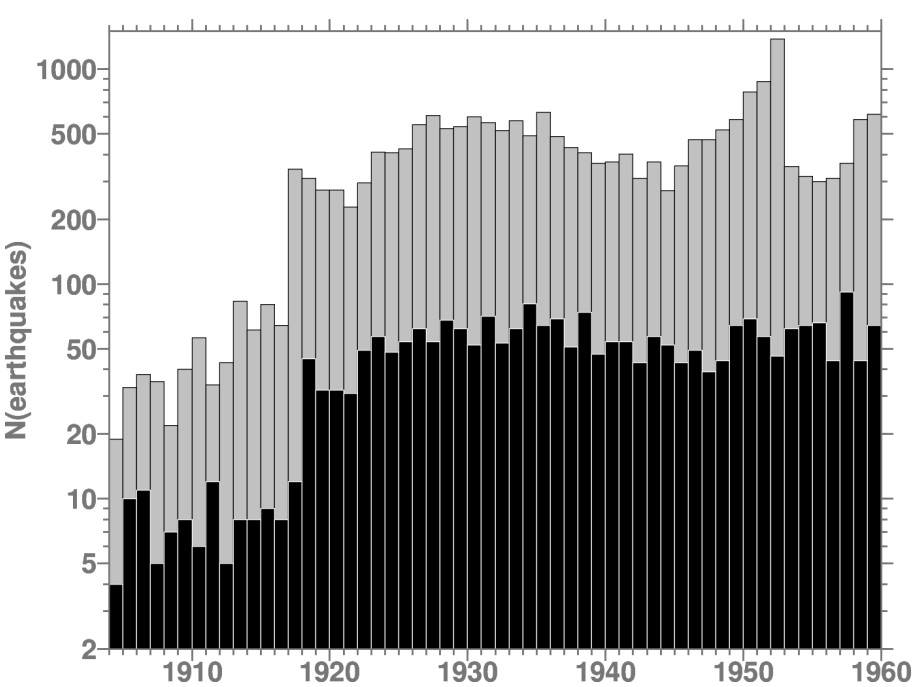

**Figure 1.** Annual number of pre-1960 earthquakes in V1 of the ISC-GEM Catalogue (black, total = 2,439 events) and the events that are available in the ISS between 1918 and 1959 and the augmented Centennial Catalogue/BAAS between 1904 and 1917 (grey, total = 20,865 events) that were not processed for V1 (in the text referred to as extension events).



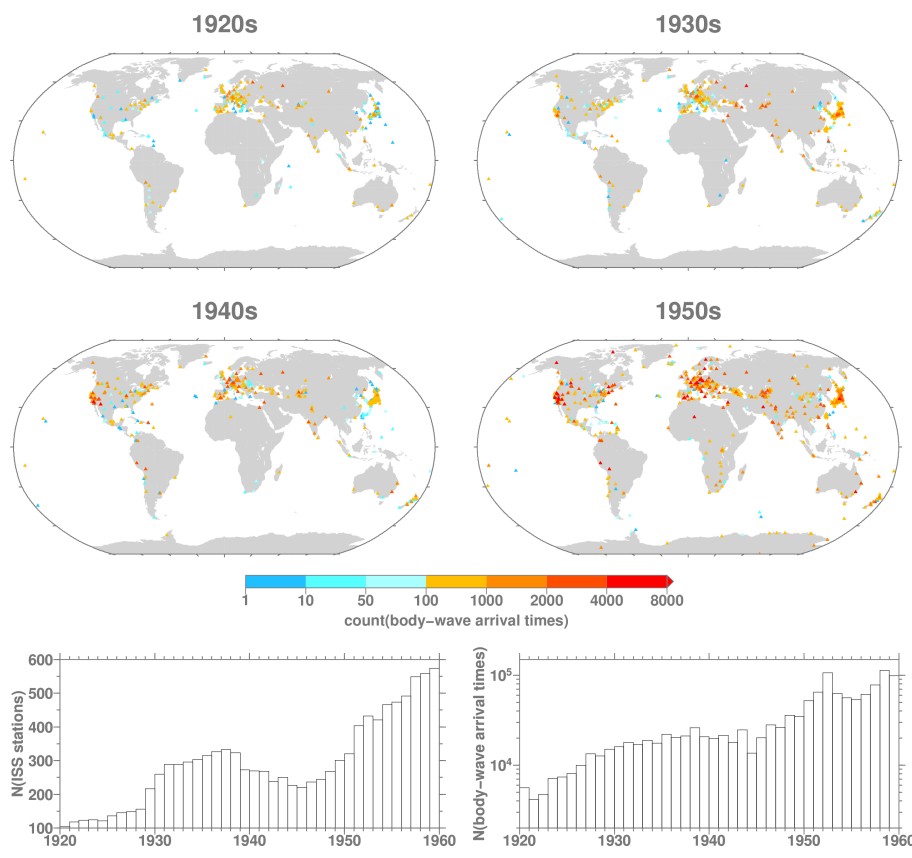

**Figure 2.** Distribution of stations listed in the ISS in each decade contributing with body-wave arrival times to the extension events and color-coded by number of body-wave arrival times. The annual number of stations (bottom left) and body-wave arrival times (bottom right) are also summarized.





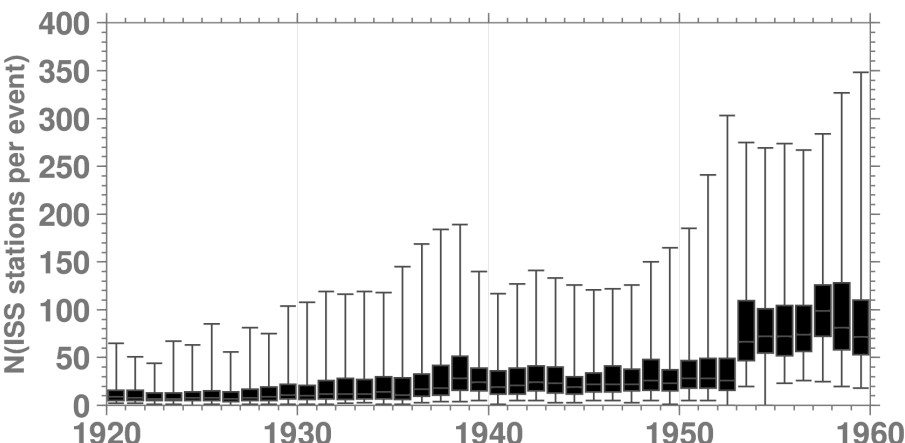

**Figure 3.** Box-and-whisker plot for the extension events of the median number of ISS stations per event in each year. The box represents the 25-75% quantile, the band inside the box the median, and the ends of the whiskers represent the minimum and maximum of all data.

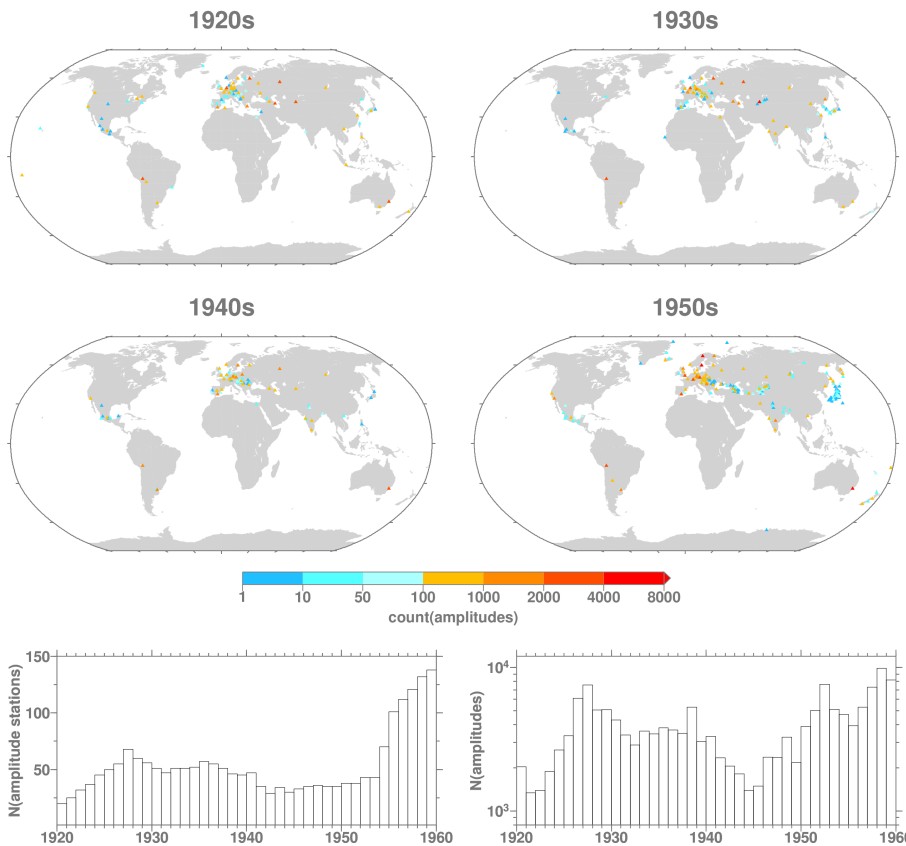

**Figure 4.** Distribution of the stations in each decade contributing amplitudes to the extension events and color-coded by number of amplitudes. The annual number of stations (bottom left) and amplitudes (bottom right) are also summarized.



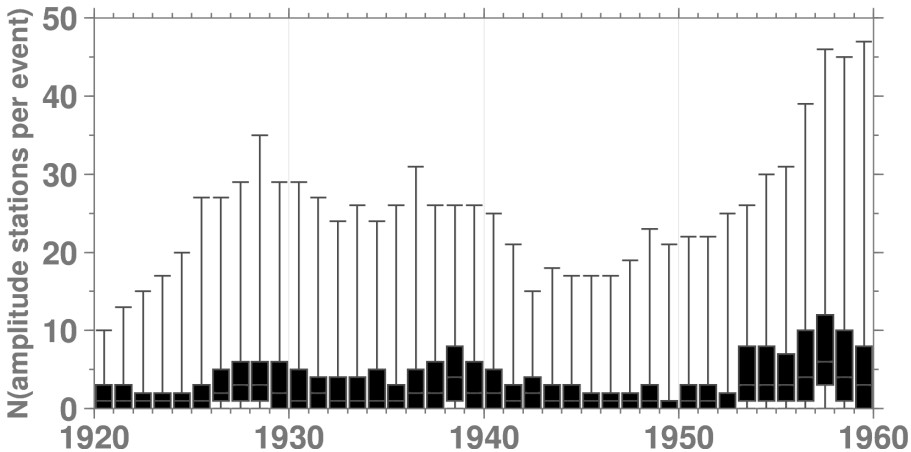

**Figure 5.** Box-and-whisker plot for the extension events of the median number of stations supplying amplitudes per event in each year. The box represents the 25-75% quantile, the band inside the box the median, and the ends of the whiskers represent the minimum and maximum of all data.

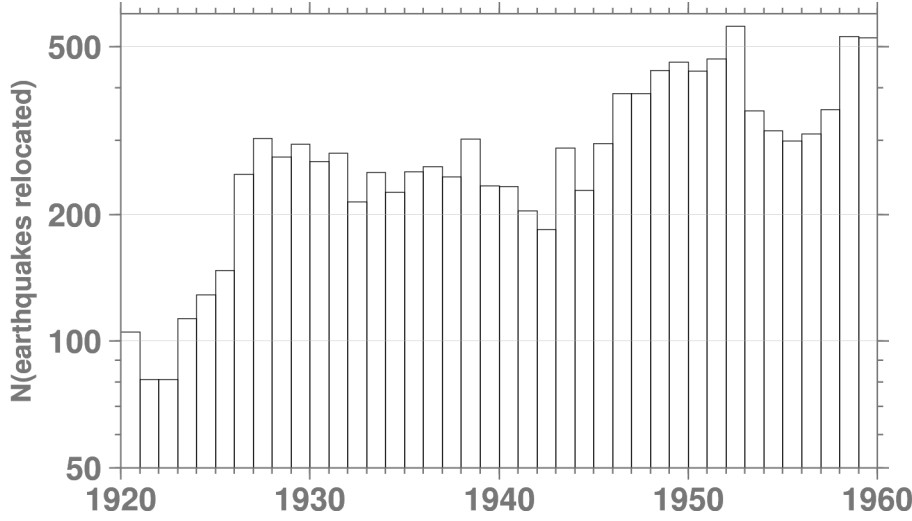

**Figure 6.** Annual number of the relocated extension events between 1920 and 1959. See text for details.



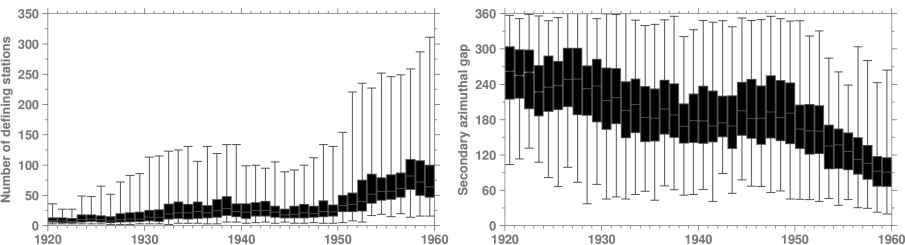

**Figure 7.** Box-and-whisker plots of the number of defining stations (NDEFSTA, left) and the secondary azimuthal gap (SGAP, right) in each year.

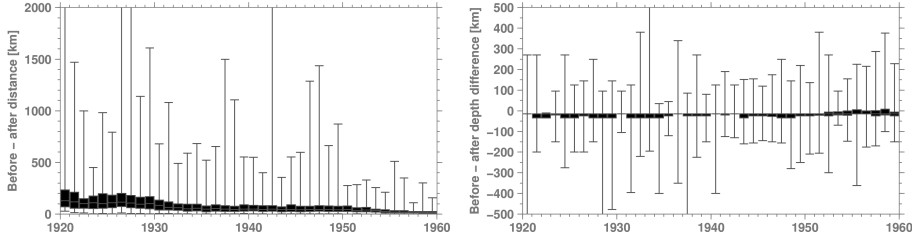

**Figure 8.** Box-and-whisker plots of the epicentre (left) and depth (right) differences between previous hypocentres (before) from ISS (or authors adopted by ISS) and ISC-GEM (after) locations in each year. For 9,417 of the 11,572 extension events relocated between 1920 and 1959 the depth for the previous hypocentres (before) was unknown and nominally set to zero.

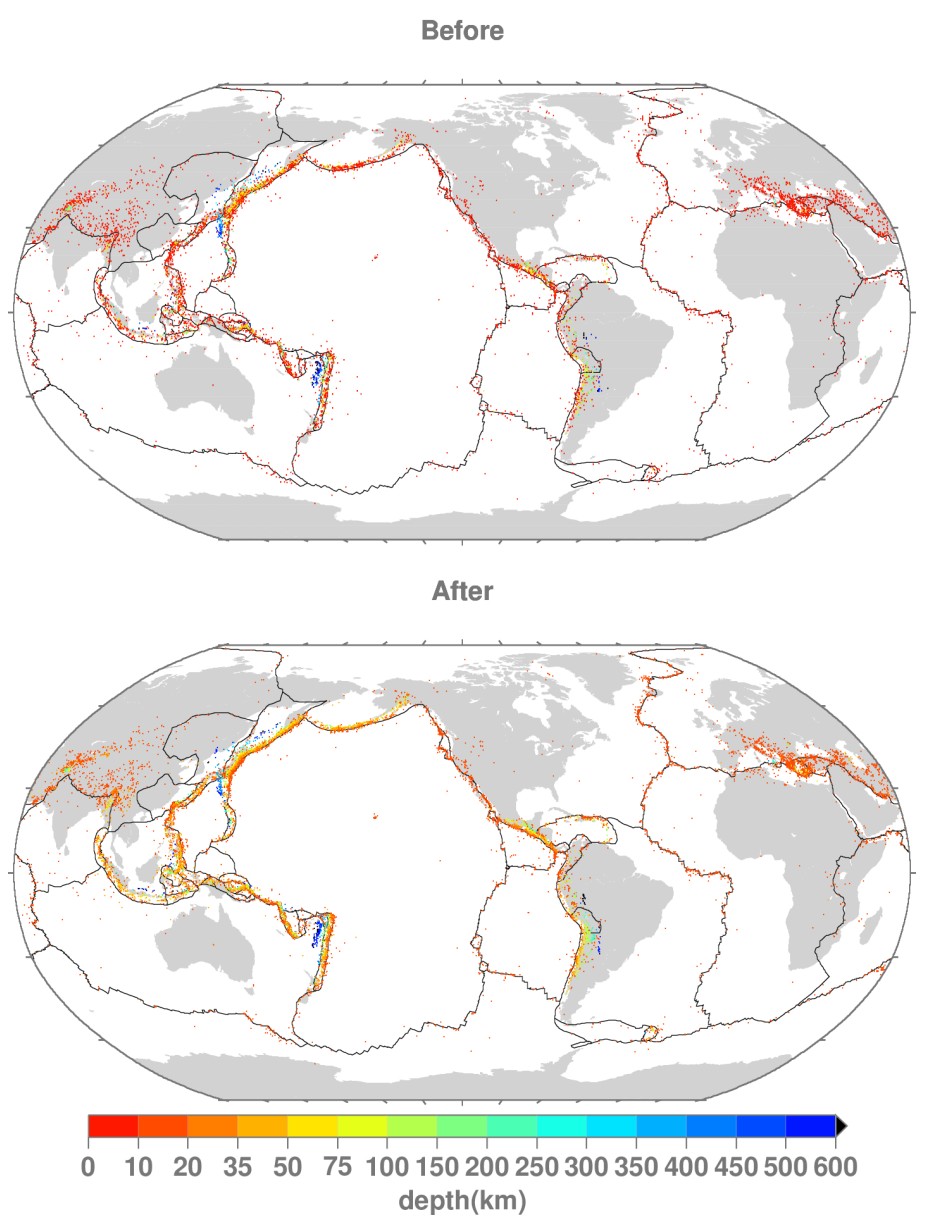

**Figure 9.** ISS (or authors adopted by ISS) (before) and ISC-GEM locations (after) for the extension events relocated between 1920 and 1959. Bird (2003) plate tectonic boundaries are also shown. It is possible to observe how the ISC-GEM locations better depict the seismicity of the Earth even on a global scale.



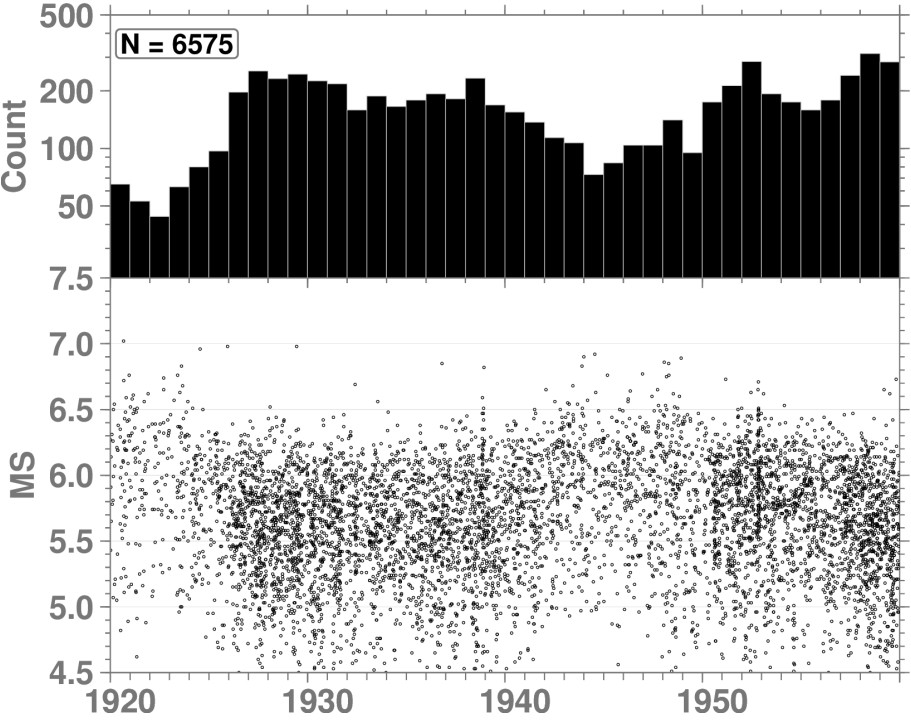

**Figure 10.** Timeline of the 6,575 re-computed MS (bottom) for the relocated extension events during 1920-1959 and their annual counts (top). 2,304 earthquakes have MS < 5.5 and 80 have MS ≥ 6.5.

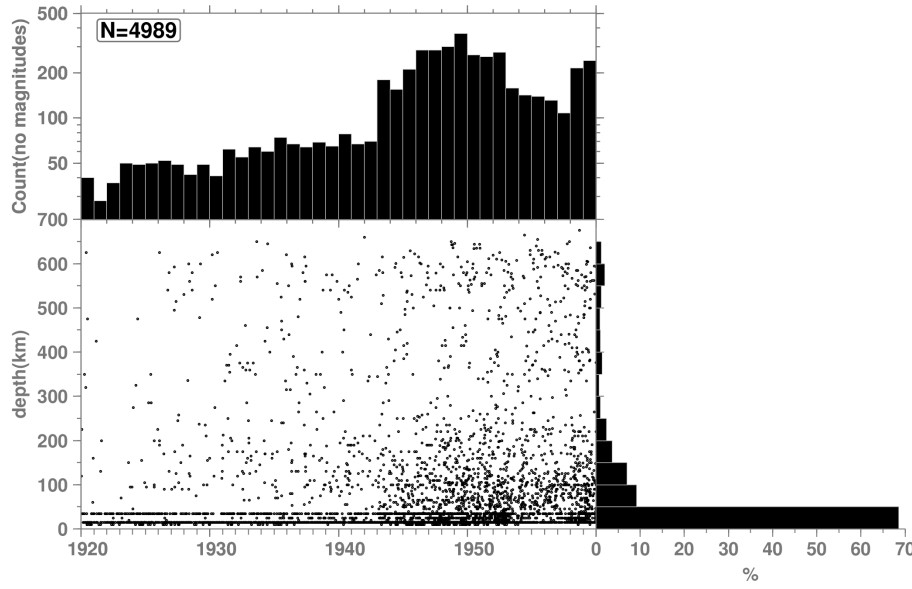

**Figure 11.** Timeline (bottom left) of the relocated extension events without magnitude between 1920 and 1959 along with their depth frequency (bottom right) and annual counts (top).



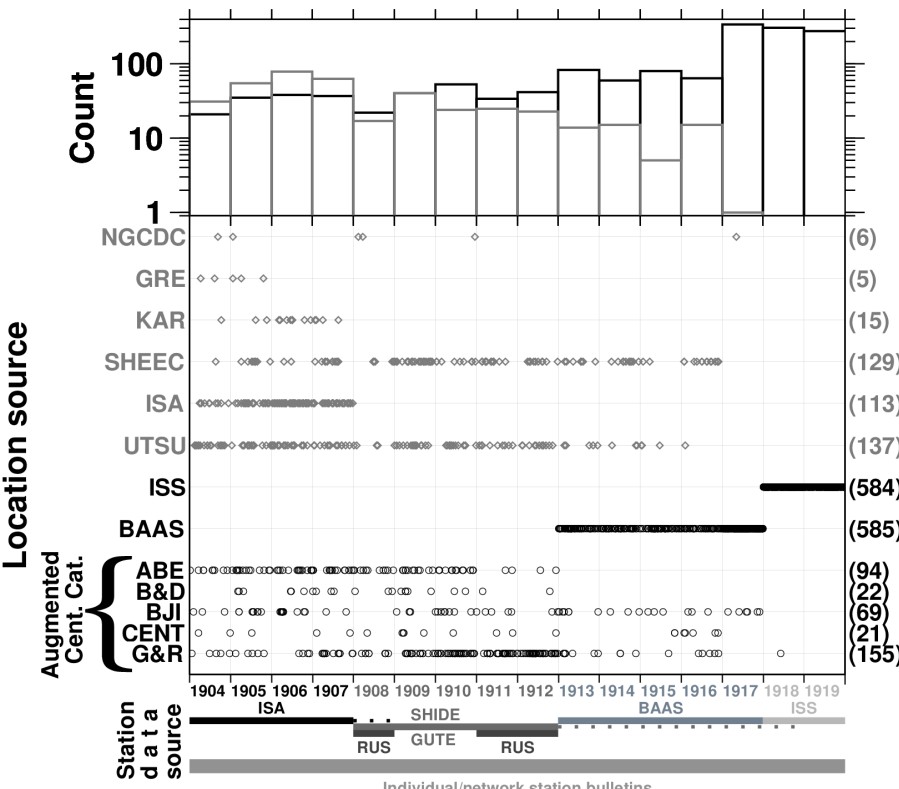

**Figure 12.** Timelines of the extension earthquakes already in our record (black circles) and added ones (grey diamonds) split by original location author/source. See text for the augmented Centennial Catalogue authors (black) and a brief descriptions of the additional location sources (grey). The total counts for each location source is shown on the right-hand side. The annual counts (top) of the extension earthquakes already known and added ones (black and grey histograms, respectively) are summarized. The station data sources (bottom) are also outlined and shown in different grey colours for the time ranges they have been used for (see text for details). For 1908 we have added station data from the "Kleinere beben" part of ISA (black dots) and during 1913-1918 we also looked into the GUTE notepads for earthquakes not listed in the BAAS and ISS (dark grey dots). Individual/network station bulletins have been used to add both surface wave amplitudes and body-wave arrival times between 1904 and 1919.

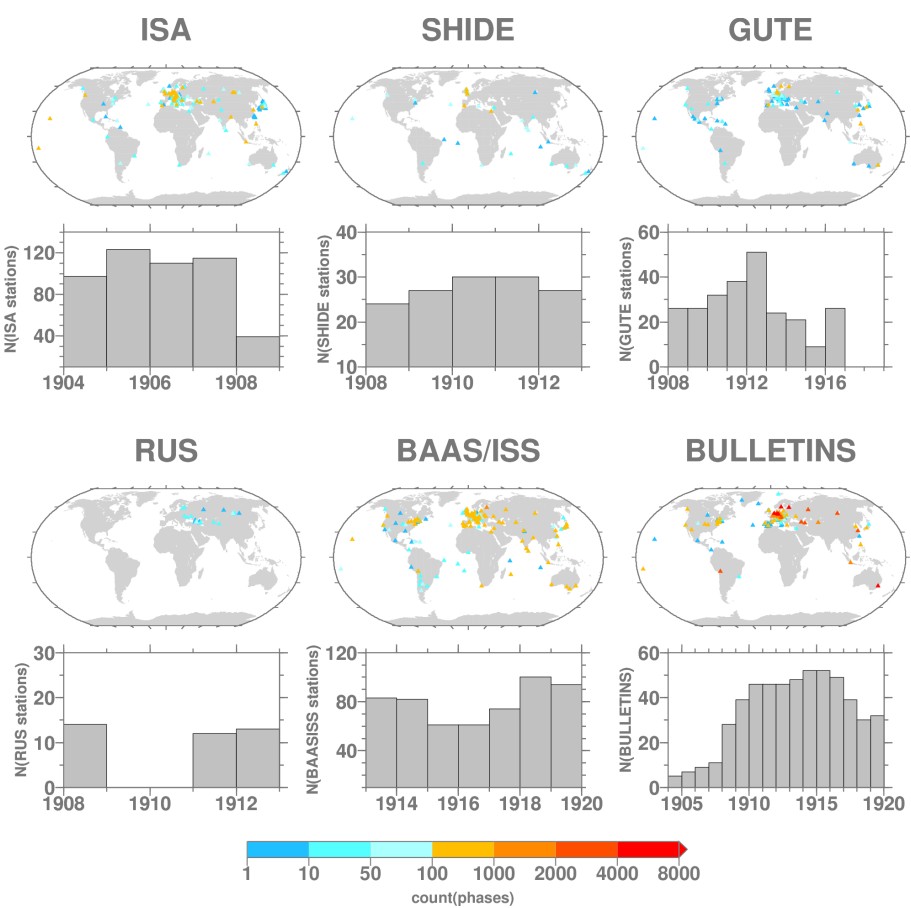

**Figure 13.** Station data contributions from ISA (top left), SHIDE (top middle), GUTE (top left), RUS (bottom left), BAAS and ISS (bottom middle) and BULLETINS (bottom right) identifying the individual/network station bulletins. The stations are colour-coded by number of phases digitized (see text for details). For each station data source, the annual station counts are shown below the corresponding map (note the different scales for each contributor).



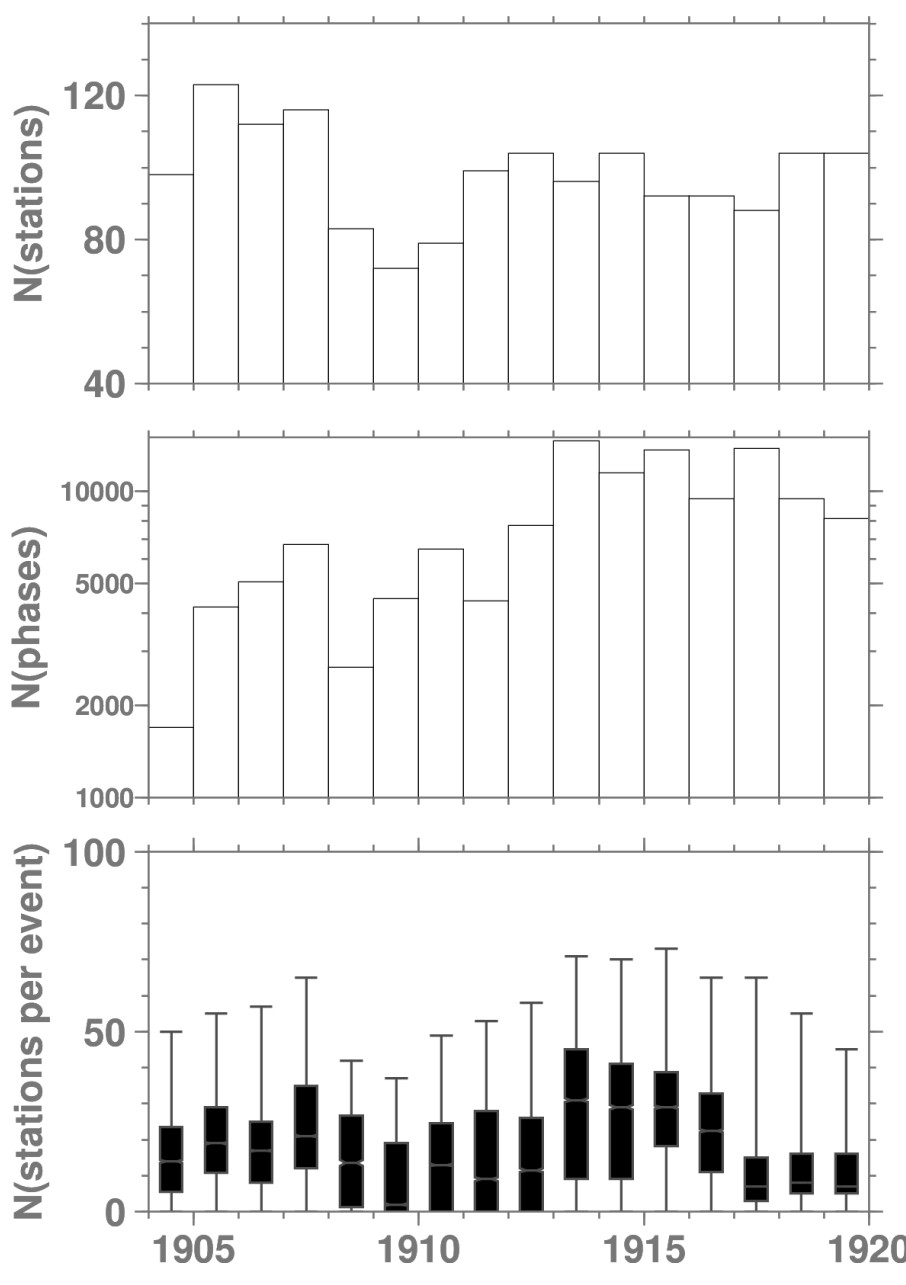

**Figure 14.** Annual number of station (top), phases (middle) and box-and-whisker plot (bottom) for the overall station data collected between 1904 and 1919.



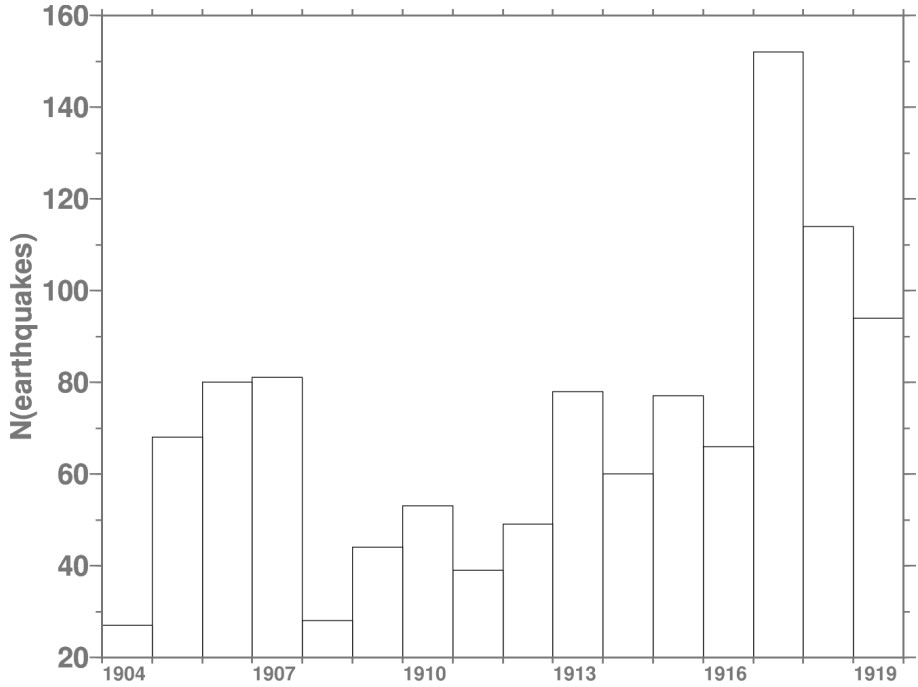

**Figure 15.** Annual number of relocated extension events between 1904 and 1919. See text for details.

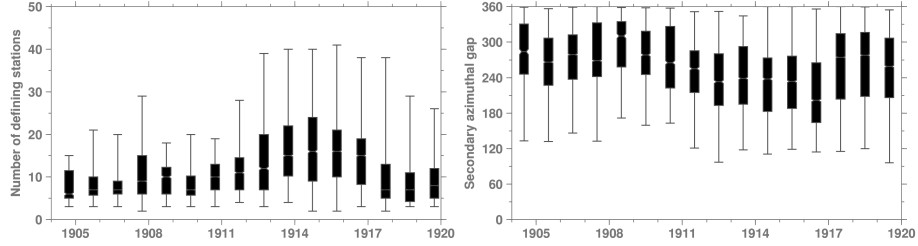

**Figure 16.** As for Fig. 7 but for the period 1904-1919.



**Figure 17.** Box-and-whisker plots of the epicentre (top), depth (middle) and origin time (OT) differences between previous (before, see Fig. 12) and ISC-GEM hypocentres (after) in each year. For 880 of the 1,110 extension events relocated between 1904 and 1919 the depth of the previous locations was unknown and nominally set to zero.



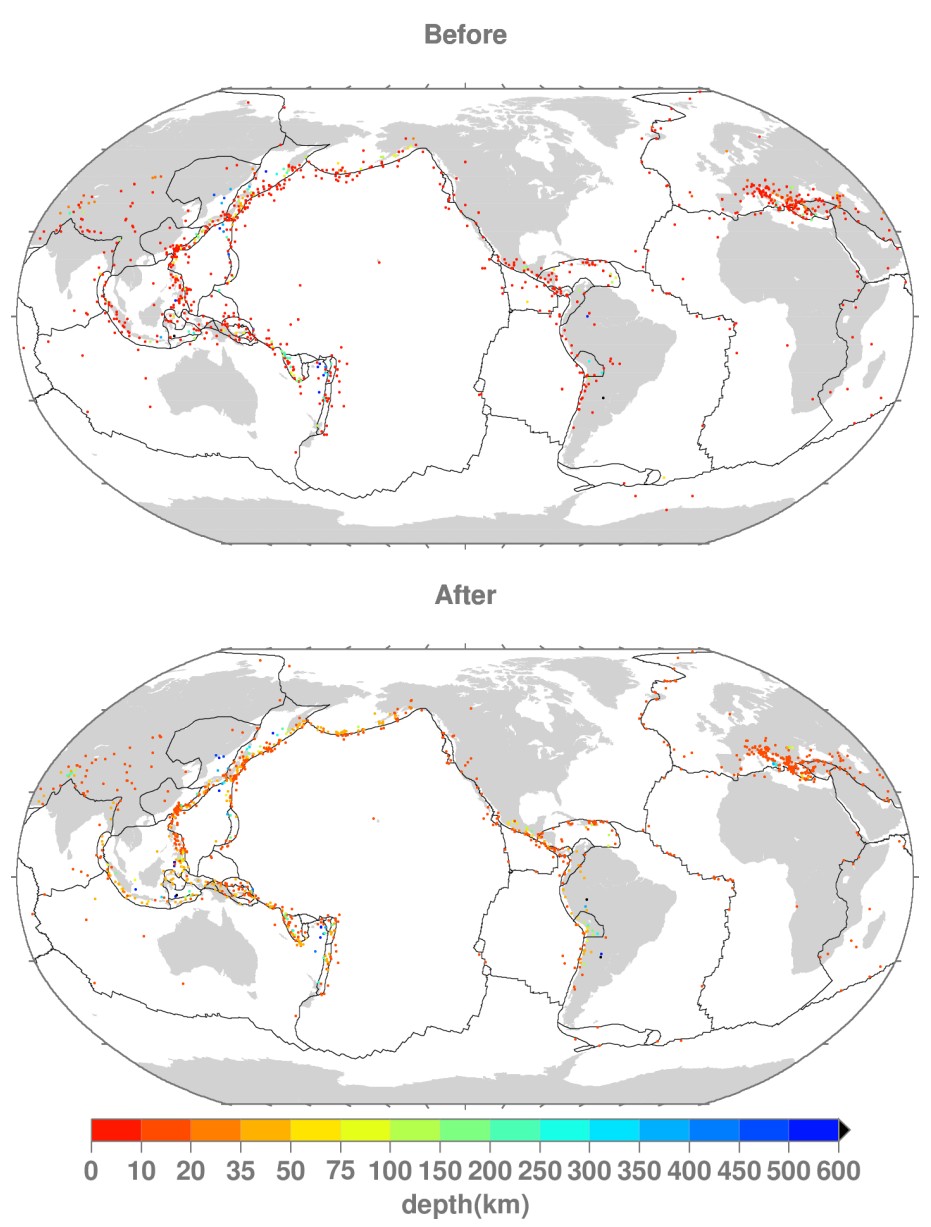

**Figure 18.** As for Fig. 9 but for 1904-1919. The location authors of the before map are outlined in Fig. 12.





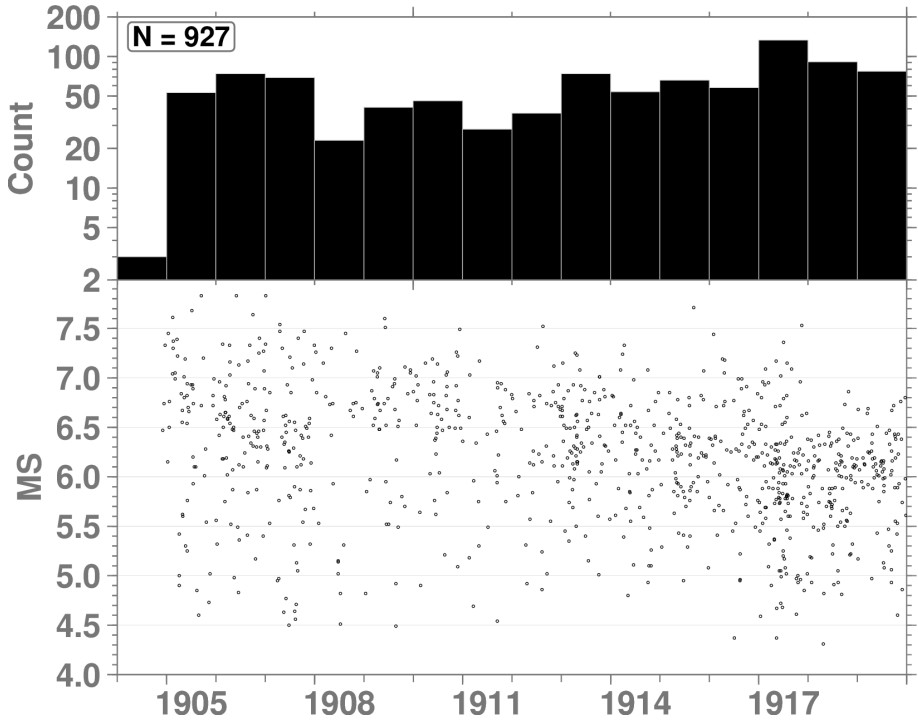

**Figure 19.** As for Fig. 10 but for 1904-1919.

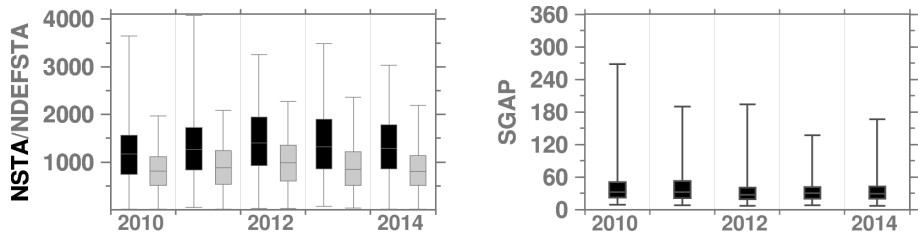

**Figure 20.** Box-and-whisker plots of (left) the number of stations (NSTA, black) and defining stations (NDEFSTA, grey) and (right) of the secondary azimuthal gap (SGAP) in each year for the earthquakes added for the period 2010-2014.

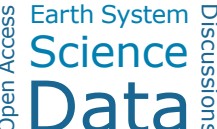

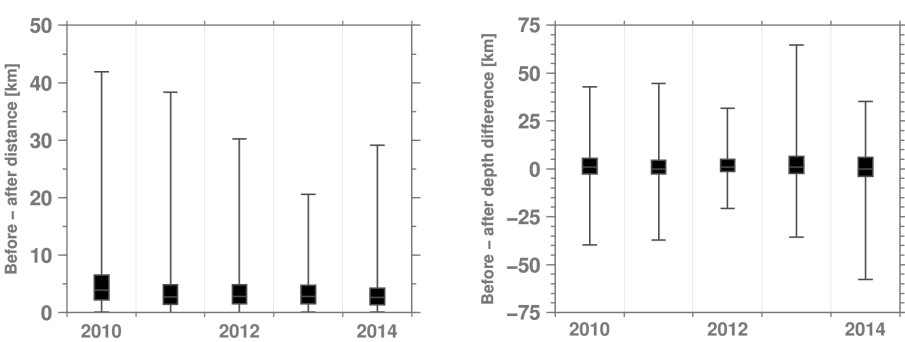

**Figure 21.** As for Fig. 8 but for 2010-2014. Note the different scales compared to Figs. 8 and 17.




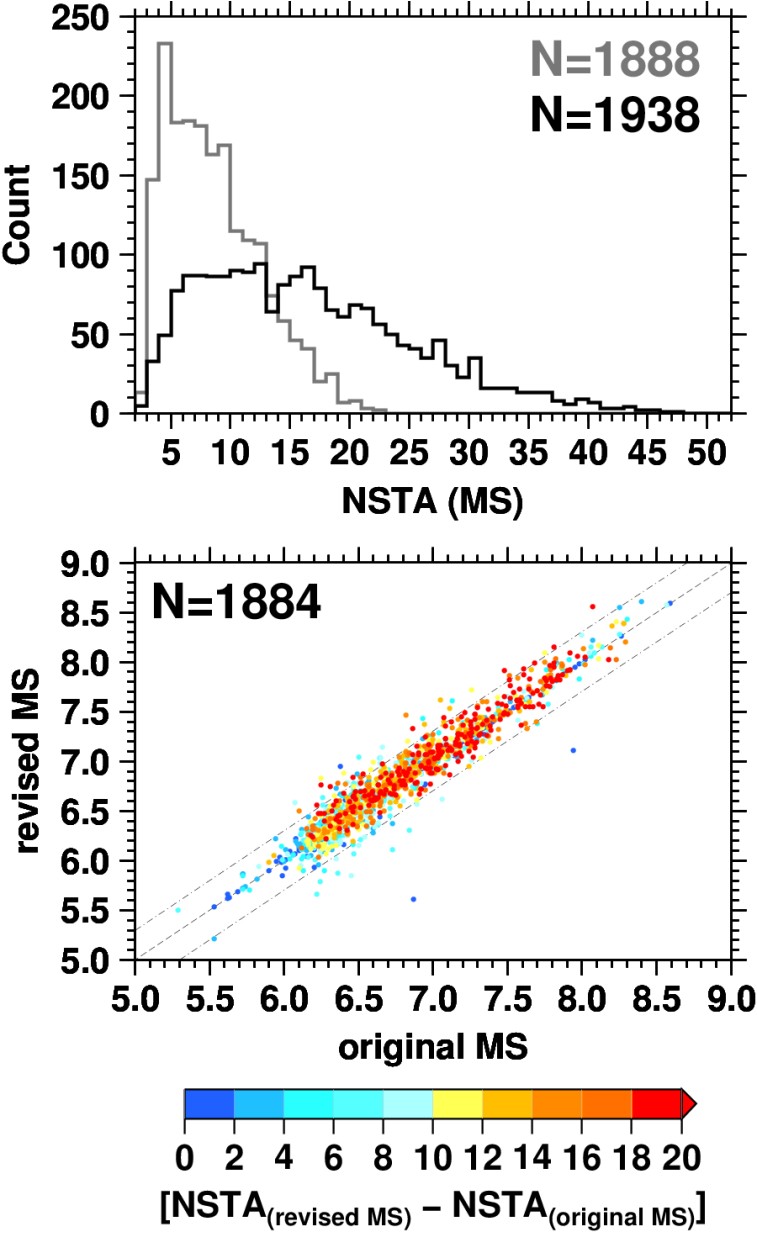

**Figure 22.** Top: histogram distributions of the number of stations (NSTA) contributing to MS for the pre-1960 earthquakes already in previous versions of the ISC-GEM Catalogue (grey) and after revision (black) using the newly added amplitude data; Bottom: Comparison between original and revised MS colour-coded by the difference of NSTA used to obtain MS. The black dashed and the dotted-dashed lines are for the 1:1 and the ±0.3 values, respectively. Note that during revision we dropped the MS for four earthquakes.





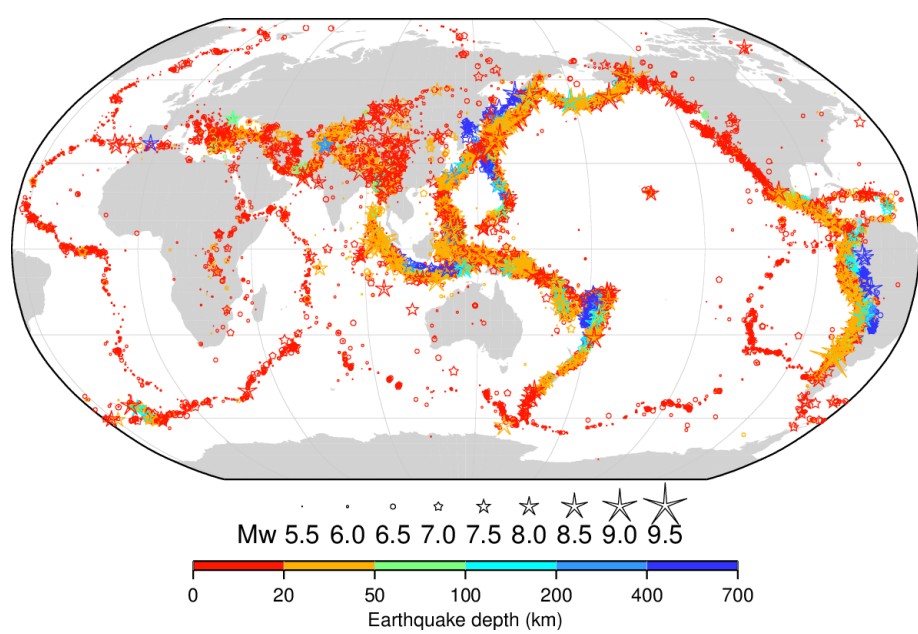

**Figure 23.** Map showing the earthquakes listed in V5 of the ISC-GEM Catalogue (more than 28,000 earthquakes, see Fig. 24). The symbols are plotted according to Agnew (2014) and colour-coded according to the ISC-GEM depth. The earthquakes shown are from the Main catalogue plus those earthquakes in the Supplementary catalogue that have reliable magnitude but poor location (i.e., magnitude quality flag not equal to D and location/depth quality flag equal to D)

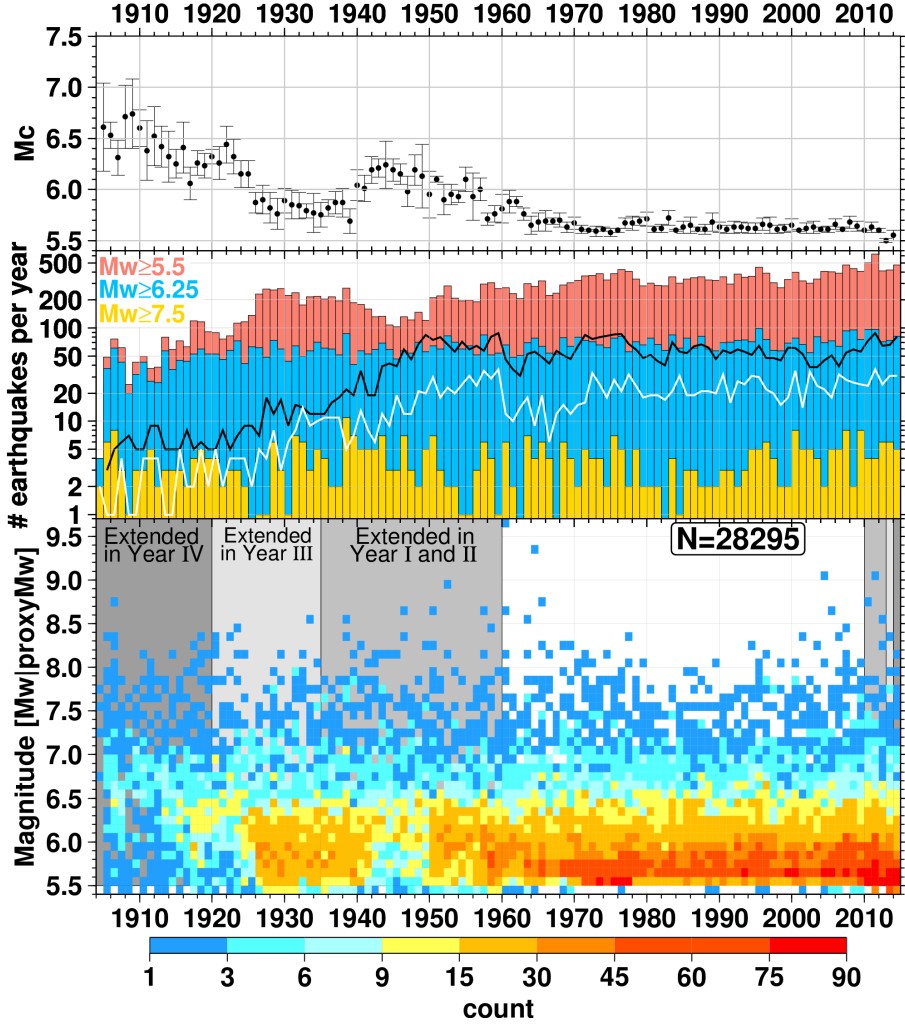

**Figure 24.** Bottom: time-magnitude distribution color coded for cells of 0.1 m.u. in each year for the earthquakes used to produce Fig. 24. Middle: cumulative annual number of earthquakes with Mw ≥ 5.5 (red), ≥ 6.5 (blue) and ≥ 7.5 (yellow) along with the annual counts of intermediate-depth (60 km < depth < 300 km, solid black line) and deep (≥ 300 km, solid white line) earthquakes; the latter two lines are obtained considering all earthquakes in V5 (i.e., both Main and Supplementary Catalogue). Top: annual completeness magnitude (Mc, black circles ±1 standard deviation) estimated with the maximum curvature method of Wiemer and Wyss (2000) implemented in the R-code of Mignan and Woessner (2012). Note that we skipped 1904 for the Mc assessment due to the small number of earthquakes.



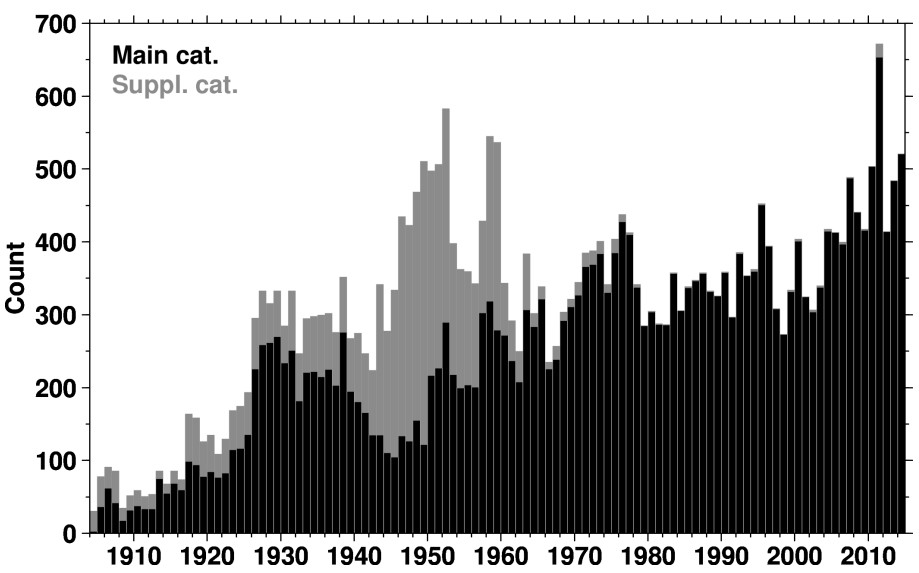

**Figure 25.** Annual number of earthquakes in the Main (black) and Supplementary (grey) ISC-GEM Catalogue; from the count of the Supplementary catalogue we have excluded a subset of earthquakes with MS below 5.



t

**Table 1.** The 4-year plan for the Extension Project (started at the end of 2013).

| Year of the Extension Project | Time period extended | Time period added |
|---|---|---|
| I | 1950-1959 | 2010-2011 |
| II | 1935-1949 | 2012 |
| III | 1920-1934 | 2013 |
| IV | 1904-1919 | 2014 |

t

**Table 2.** Criteria to assign the location quality flags for location, depth and magnitude. SGAP is the secondary azimuthal gap, GTCAND denotes a high confidence location accuracy that makes the event a candidate for the IASPEI Refence List (Bondár and McLaughlin, 2009, see also www.isc.ac.uk/gtevents), depdp is the depth constrained by depth phases (if available), NSTA10 is the number of stations within 10 km and NSTA(local) is the number of stations within 150 km (Bondár and Storchak, 2011). MS is considered well constrained when it is obtained from more than 4 stations, within 5.5-7.5, and has uncertainty $\leq 0.2$. See text for details.

| Quality flag | Location | Depth | Magnitude (Mw, direct or proxy) |
|---|---|---|---|
| A | SGAP < 120 && eccentricity < 0.75 or GTCAND | depdp or GTCAND or NSTA10 | GCMT |
| B | SGAP < 160 | NSTAlocal > 2 | Literature or Proxy based on well constrained MS |
| C | Other cases | Other cases | Literature or Proxy based on poorly constrained MS or Proxy based on mb |
| D | Manually assigned | Manually assigned | No magnitude or Proxy uncertainty > 0.7 or Mw proxy based on MS < 5 |



t

**Table 3.** Summary of the location and depth quality flags for the extension events between 1920 and 1959.

| Quality flag | Count for location | Count for depth |
|:---:|:---:|:---:|
| A | 1744 | 1886 |
| B | 3315 | 479 |
| C | 6431 | 9208 |
| D | 83 | 0 |

t

**Table 4.** Summary of the magnitude quality flags for the relocated extension events between 1920 and 1959. Included in the D flag are 4984 events where no magnitude was recomputed.

| Quality flag | Count for magnitude |
|:---:|:---:|
| A | 0 |
| B | 3030 |
| C | 2824 |
| D | 5719 |

t

**Table 5.** Summary of the location and depth quality flags for the relocated extension events between 1904 and 1919.

| Quality flag | Count for location | Count for depth |
|:---:|:---:|:---:|
| A | 14 | 6 |
| B | 75 | 2 |
| C | 775 | 1102 |
| D | 246 | 0 |

t

**Table 6.** As for Table 4 but for 1904-1919. Included in the D flag are 183 events where no magnitude was recomputed.

| Quality flag | Count for magnitude |
|:---:|:---:|
| A | 0 |
| B | 420 |
| C | 427 |
| D | 262 |



t

**Table 7.** Number of earthquakes added between 2010 and 2014.

| Year | Count |
|------|-------|
| 2010 | 504 |
| 2011 | 672 |
| 2012 | 414 |
| 2013 | 484 |
| 2014 | 521 |