# Peer review of "The ISC-GEM Earthquake Catalogue (1904-2014): status after the Extension Project"

_Earth System Science Data, 2018_

## Referee Comment (RC1) · Anonymous Referee #1 · 28 Aug 2018

General Comments

The extended ISC-GEM catalogue of earthquakes and associated commentary on its development are both critical resources to the earth science community. As the authors note, a list of earthquakes with the longest possible temporal duration and most consistent possible physical criteria is requisite for a range of research endeavors, including identification of temporal and spatial patterns, testing of various hypotheses related to formation of such patterns, and development of potential actuarial or theoretical tools for earthquake forecasting. The longer the duration of such a catalog, the more effective are statistical approaches. The more consistent the catalog and realistic the uncertainties, the more effective are hypothesis tests.

As the authors note, previous attempts to create long earthquake catalogs, including

the most common: PAGERCAT, the Centennial Catalog, and the USGS/NEIC, rely on the compilation of event information from multiple sources. As a result, these previously developed catalogs are very likely to be internally inconsistent with respect to magnitude estimates and uncertainty thereof, as well as with respect to depth and epicentral location (to a lesser extent). Such biases and uncertainties alias into all efforts to answer key questions about global earthquakes, such as whether moment release rate varies in time and space, the extent to which quakes can directly trigger one another, and which forcings may be coupled to earthquake cycles. For example, several recent publications using different earthquake catalogs and different statistical methods differ on whether the largest quakes are clustered in time or not (e.g. Abe and Suzuki, 2004; Ben-Naim et al., 2013; Beroza, 2012; Bragato and Sugan, 2014; Daub et al., 2015; Michael, 2011; Sammis and Smith, 2013; Shearer and Stark, 2012) and whether it may be possible or not to forecast large events with time-dependent probabilities based either on past earthquake occurrence or other potential physical forcing (e.g. Barbot et al., 2012; Bendick and Bilham, 2017; Dieterich and Richards-Dinger, 2010; Zahn and Shearer, 2015). All such work is critically sensitive to catalog quality. Consequently, future related and novel efforts will benefit from the initial development of the ISC-GEM catalog and its recent extensions.

The most notable updates to the catalog consist of relatively large numbers of added events in the early 20th century. This is the part of previous catalogs that is most inconsistent, so such careful additions have the greatest impact. Many of the events are derived from paper records, journals, and other obscure sources which are not readily available and do not have consistent or well-known quality, so they especially benefit from the careful re-analysis required for inclusion in ISC-GEM. The extension also includes the most recent events. Although the latter are accessible from many different sources, their treatment consistent with the body of the catalog enforces the standardization practice that ensures minimum bias.

In general, the ongoing development and improvement of a standardized earthquake

catalog with clearly reported methods and uncertainties is an incredibly valuable scientific contribution. Widespread uptake of such a catalog, even if it has some flaws and missing records, will mean that a wide range of investigations of seismicity patterns can be compared and integrated into a broader understanding of earthquakes, whether their fundamental physical dynamics or applied statistics.

Technical Comments

There are two big considerations embedded within the extended ISC-GEM catalog. They both likely arise from the requisite consistent treatment of observational data over time. First, strictly using the latest extended ISC-GEM catalog would imply that earthquake productivity has systematically increased through the 20th century (Figure 1). This is certainly due to the strict criteria for inclusion used by the ISC-GEM group, which results in the rejection of many more events prior to the 1940s than later, and is noted and acknowledged in the methodological description. An alternative approach, apparently taken by the PAGERCAT group, is to relax the selection criteria in the early century when the observational data are poor. With this strategy, the global seismic productivity does not have a strong secular trend (or it is at least not statistically significant). The difference in selection strategy appears in the figures below as a decreasing difference between the two catalogs, which pretty much goes away between 1950 and 1960 (Figure 2).

Second, the reported uncertainties on ISC-GEM events are very large in the early 20th century, making it difficult to assess the catalog completeness at any specific magnitude threshold (Figure 3).

Both of these effects arise from enforcing a consistent treatment of event observations for the entire catalog duration, and it is not clear that there is any better way to handle the differences in data quality over time. However, these effects also make it rather difficult to use the full catalog, as they have the potential to hide real signals under selection limits and magnitude uncertainties. It might be useful for the commentary to

point out that, in addition to the acknowledged likelihood of missing events early in the century, these missing events alias into an unlikely-to-be-real trend in number of large events and therefore seismic productivity. Alternatively, the commentary could include a short additional section that specifically outlines the implications of the event counts (as, for example, clearly shown in figure 1) to hypotheses about temporal patterns. Some note of how the ISC-GEM compares to other commonly used catalogs should be included in the same section. Such a commentary would help ensure that researchers using the catalog are sufficiently aware of its inevitable biases. As long as studies using the ISC-GEM catalog either acknowledge the resulting systematic bias or use additional events from other catalogs to "fill-in", the impacts can be addressed, if not mitigated. However, any studies using the whole catalog without consideration of the selection bias may produce spurious results.

Figure 1: Counts of annual earthquakes above indicated magnitude thresholds in Mw=0.2 increments for the ISC-GEM catalog (5-year averaging). For comparison purposes, the dashed line represents the annual Mw$\geq$7 seismicity rate from Pagercat (average 15.1$\pm$2.9/yr, reducing by 2.9 earthquakes per century).The ISC-GEM Mw$\geq$7 count (11.1$\pm$2.9/yr) increases by 4.9 earthquakes per century.

Figure 2: The difference (solid green) between Pagercat (red) and ISCGEM (black) Mw$\geq$7earthquakes (and the smaller difference between the raw ISC/GEM catalog and the mean of 20 ISCGEM synthetic catalogs (dashed green line).

Figure 3: Example showing the rate of Mw$\geq$6.6 earthquake productivity if Mw uncertainties are included. The blue line is the raw ISCGEM catalog and red line the smoothed 5 year running mean of 20 synthetic ISCGEM catalogs (grey). Green dots indicate $\pm$ ISCGEM uncertainties for individual earthquakes (right hand scale in Mw). The orange line is a 5 year gaussian smoothing estimate.
* * *
[Figure]

**Fig. 1.** Quake counts (see text for full caption)

[Figure]

**Fig. 2.** Comparison with PAGERCAT (see text for full caption)

[Figure]

**Fig. 3.** Magnitude uncertainty (see text for full caption)

---

## Referee Comment (RC2) · Anonymous Referee #2 · 12 Sep 2018

The authors update and extend the ISC-GEM catalogue, which is a reference global catalog developed mostly for seismic hazard assessment purposes, but which is also widely used for other studies, including seismo-tectonics. The main difference with respect to the previous version of the ISC-GEM catalog (v1) is the inclusion of pre-1960 earthquakes with magnitudes below the original (time-dependent) magnitude thresholds and new M>=5.5 events from 2010 to 2014, as well as a better assessment of magnitudes for some earthquakes.

Because the methodology is sound and has been presented in earlier articles, I will not comment much on it. Most of my comments, which are detailed below, concern presentation issues.

In particular, I find that the reference throughout the article to the 4-year project and

[Figure]

its stages is somewhat cumbersome and uninteresting for the reader. While such references would be adequate in a technical project report, readers of the article will be interested in the catalog and not on how the project was timed and developed. I thus encourage the authors to eliminate from the article references to the yearly development of the extension project, and rather focus only on the catalogue analysis.

I also find a bit difficult to follow the description of the data and workflow. It would help if the authors could add a flowchart with the processing. The dataset could be identified at each step as extension-0, extension-1, extension-x, extension-final, and the processing steps could be identified as well. A table could accompany the flowchart, including the number of events after each processing step. If needed, adapt for the different time periods.

The article is written in easily understandable English, but still some sentences require re-reading. If possible, I would recommend that the article be carefully edited for English.

The article is pertinent, of general interest, and relevant for all users of the catalog, and well as for readers interested in the development of seismic catalogues. I therefore recommend the article for publication after minor revision.

Minor comments:

. Page 1, Lines 1-5: Too long sentence. Becomes confusing. Rephrase/clarify.

. P1, L 11-13: Include a brief explanation as to why the ISC-GEM catalog now starts only in 1904 rather than 1900 (as originally).

. P2, L7: "large earthquakes pre-1964 and from 1964 onwards,"- confusing, rephrase.

. P3, L24. I would suggest to remove the text and table (Table 1) about the timing of the project, unless there is a good reason to show it.

. P5, L3: Remove the reference to the phase of the project, leave only the reference to

catalogue time period (1920-1959).

. P 5, L12-14. Confusing. Rephrase/clarify.

. P6, L7-12. The criteria are somewhat vague. Is it possible to detail, for example in a table?

. P6, L18-19. Confusing. Rephrase/clarify.

. P7, L5. Replace "recurrent" by "frequent".

. P9, L16, and later. I find it confusing to use the word "counts" to refer to the number of locations available in each source. Maybe "reported locations" or some different word/expression is more adequate. "Counts" seems too generic.

. P11, L28-29. Rephrase/clarify.

. P14, L22. Careful with referring to "historical" earthquakes, as this expression is typically used for earthquakes documented historically but not recorded instrumentally. Maybe better to use "pre-digital", if this is the case?

. Figure 1. Identify in the figure the different time periods referred to in the text (P3, L14).

. Figure 6. It would be nice to plot, behind this histogram, a second histogram showing the initial number of extension events in the period studied.

. Figure 23. It would be interesting to add to this figure a similar one, but showing only the earthquakes with well-constrained locations.

. Acknowledgements: Josep Batlló, there is no "h" at the end of "Josep", and there are two "l" and one "t" in "Batlló" (!).

Finally, I apologize the editor and authors for the late review.
* * *
[Figure]

2018.

---

## Author Comment (AC1) · 26 Sep 2018

**Author's Response to the Reviewer's Comments #1 (PAPER: The ISC-GEM Earthquake Catalogue (1904-2014): status after the Extension Project, https://doi.org/10.5194/essd-2018-59)**

Domenico Di Giacomo          E. Robert Engdahl          Dmitry A. Storchak

September 26, 2018

**Response to general comments**

We thank the reviewer for recognizing the extended ISC-GEM Earthquake Catalogue as a "critical resource to the earth science community" and for all other positive feedback. We also thank the reviewer for reminding us of PAGER-CAT, which we now include in the **Introduction** (in the revised version we have added the reference Allen et al. (2009) after the first mention of the Centennial Catalogue). Since specific points are not raised in the general comments, here we make only a brief remark and provide detailed answers to the reviewer's technical comments.

Throughout this document the Reviewer Comment (RC) are reproduced in bold and the Author Response (AR) in italic.

**RC: As the authors note, previous attempts to create long earthquake catalogs, including the most common: PAGER-CAT, the Centennial Catalog, and the USGS/NEIC, rely on the compilation of event information from multiple sources. As a result, these previously developed catalogs are very likely to be internally inconsistent with respect to magnitude estimates and uncertainty thereof, as well as with respect to depth and epicentral location (to a lesser extent).**

*AR: Indeed, one of the main motivations behind the production of the ISC-GEM Catalogue has been the necessity to homogenise (to the largest extent possible) the location and magnitude parameters of global earthquakes instrumentally recorded and to provide formal uncertainties. However, to our knowledge, none of the previous most common catalogues provide magnitude and location uncertainties for earthquakes that occurred before the modern era (starting approximately in the early 1960s). Furthermore, the inconsistencies of previous catalogues in terms of location are not less significant than for magnitude. This is shown, for example, in Bondár et al. (PEPI 2015) as well as in Fig. 9 and 18 of the submitted paper.*

**Response to technical comments**

**RC: There are two big considerations embedded within the extended ISC-GEM catalog. They both likely arise from the requisite consistent treatment of observational data over time. First, strictly using the latest extended ISC-GEM catalog would imply that earthquake productivity has systematically increased through the 20th century (Figure 1). This is certainly due to the strict criteria for inclusion used by the ISC-GEM group, which results in the rejection of many more events prior to the 1940s than later, and is noted and acknowledged in the methodological description. An alternative approach, apparently taken by the PAGER-CAT group, is to relax the selection criteria in the early century when the observational data are poor. With this strategy, the global seismic productivity does not have a strong secular trend (or it is at least not statistically significant). The difference in selection strategy appears in the figures below as a decreasing difference between the two catalogs, which pretty much goes away between 1950 and 1960 (Figure 2).**

*AR: we appreciate the reviewer's comment and before addressing it specifically, we feel that we have to reiterate why it is important we stick to our approach and why the ISC-GEM Catalogue is different from other catalogues compiled by simply merging different sources (e.g., PAGER-CAT). We do this first by considering a particular earthquake and then by showing the comparison of our re-computed MS with four different past catalogues. Such examples are to support our arguments and hopefully better highlight how the ISC-GEM Catalogue is different from the most common earthquake catalogues (including PAGER-CAT).*

*In order for an earthquake to be listed in the main file of the ISC-GEM Catalogue we require that sufficient instrumental data has been gathered to allow us to reassess location and magnitude in a reliable way (location and magnitude flags are explained in the main text). Thus, if the data gathered for an event is scant or circumstances do not allow us to reach a sufficient quality in location and/or magnitude, then we list the event in the supplementary catalogue. In addition, as mentioned in the main text, earthquakes not selected for processing have either insufficient station data or no station data at all. The latter case is of particular importance since it poses a question mark on the validity of the earthquake's occurrence. Indeed, particularly for pre-1920 earthquakes, the lack of station data may be simply due to the fact that the earthquake as reported in a previous source either did not happen or has some gross error in one or more parameters (e.g., wrong origin time/location and/or magnitude). For example, PAGER-CAT (as well as Centennial Catalogue and other earthquake catalogue repositories such as NGCDC and UTSU mentioned in the main text) lists two earthquakes the 12 December 1908 (excerpts from PAGER-CAT):*

*#dateStr, lat ,lon ,depth, locSrc, prefMag, prefMagType, prefMagSrc*
*1908-12-12 12:08:00,-14.000,-78.000,60, B&D, 8.2, UK, B&D*
*1908-12-12 12:54:54, 26.500, 97.000, 0, G&R, 7.0, Ms, AN2*

*The first event was originally listed in Bath&Duda (locSrc=B&D) catalogue and the second in Gutenberg&Richter (locSrc=G&R). However, whilst searching station data for both earthquakes we found that no observatory in the world had recorded the first earthquake, whereas the second one is very well documented. Although at the beginning of the last century global seismological observatory practice was at its inception, it is very unlikely that a magnitude 8.2 earthquake would not be*

recorded in 1908. Nevertheless, original sources have been checked and we are confident in our conclusion that the first event was not real. The result is that the second event is listed in the ISC-GEM Catalogue (evid = 16958007), whilst the first is not. Although this case may be an extreme example, there is a more general reason for the ISC-GEM Catalogue to re-assess the magnitude (and location, but we focus on magnitude only here to answer the RC) of pre-1960 earthquakes: as already shown by Di Giacomo et al. (PEPI 2015a, Figs. 5 and 6), the quality of the magnitudes listed in previous catalogues depends on the source and are available in different time periods. Here in Figs. AR1 and AR2 we expand Figs. 5 and 6 of Di Giacomo et al. (PEPI 2015a) by including the earthquakes added during the Extension Project and including also B&D and Rothé catalogues. With the exception of Abe's catalogue, the other three catalogues show a general overestimation of about 0.2 magnitude units (m.u.) compared to our re-computed MS. This observation is not new but it is important here to further emphasize the shortcomings of the magnitudes listed in previous catalogues. Note that such catalogues are also some of the sources used to compile PAGER-CAT.

These comparisons and the case of the 1908-12-12 12:08 earthquake are to remind catalogue users of the limitations of past catalogues (particularly for pre-1960 earthquakes) and of being cautious when taking such solutions at face value (as it is too often common practice).

The results shown in Figs. AR1 and AR2 are also important to answer the reviewer's Figure 1 and 2 and the comment **'the latest extended ISC-GEM catalog would imply that earthquake productivity has systematically increased through the 20th century'**. First we want to point out that annual counts of earthquakes is not the best way to discuss the **"earthquake productivity"** derived from a catalogue. Moment release is more significant, as shown, e.g., in Figure 21 of Di Giacomo et al. (2015a). Secondly, in the reviewer's Figure 1 and 2 the curves related to the ISC-GEM Catalogue seem to start in 1902 and 1900, respectively, instead of 1904, the first year of the ISC-GEM Catalogue (Version 5). Such plots should start in 1905 as 1904 has a small number of earthquakes (see Figure 24 caption). Finally, we are not sure how the PAGER-CAT Mw curve in the reviewer's Figure 1 was created, since from https://earthquake.usgs.gov/static/lfs/data/pager/catalogs/ the preferred magnitude in PAGER-CAT is a highly heterogeneous mix of types and authors. Leaving such details aside, if we consider the curves for $Mw/M \geq 7.0$ in PAGER-CAT in Figure 1 and 2, it seems that the **"earthquake productivity"** is larger before the 1950s-1960s (this would imply that the **"earthquake productivity"** has decreased since then, according to PAGER-CAT). Such an observation supports the results shown in Figs. AR1 and AR2 as it would explain the apparent overestimation in the PAGER-CAT magnitude composition before the 1960s (at least for magnitude 7 an above). Instead, the solid black line for the ISC-GEM Catalogue shows a more significant dip up to the late 1920s. This is not surprising and we warn the reader about such a feature in the main text. However, to address the reviewer's concern, we modified the text to further stress the danger of obtaining what the reviewer called **"an unlikely-to-be-real trend in number of large events and therefore earthquake productivity"**. The changes to the text to address this and the following RCs are reproduced later in this document and highlighted in the revised version of the manuscript.

**RC: Second, the reported uncertainties on ISC-GEM events are very large in the early 20th century, making it difficult to assess the catalog completeness at any specific magnitude threshold (Figure 3).**

*AR: As mentioned earlier, to our knowledge we are the first to list systematically in an earthquake catalogue both location and magnitude uncertainties for pre-1976 earthquakes. Specifically for magnitude uncertainty, one has to consider that for pre-1976 earthquakes the magnitude uncertainty cannot be smaller then 0.2 m.u. (with the exception of GCMT solutions for deep earthquakes in 1962-1975). Indeed, the smallest magnitude uncertainty we allow for any earthquake magnitude is 0.1, meaning that, in the best case scenario, Mw proxies will have an uncertainty of 0.2 m.u. However, when the basis for Mw computation (i.e., our re-computed MS or mb) has an uncertainty larger than 0.3 m.u, the resulting Mw uncertainty will be even higher (the MS/mb uncertainty is mapped on the non-linear conversion relationship by Di Giacomo et al., 2015a). For direct Mw from the literature we assigned uncertainty normally between 0.2 and 0.4 (see Lee and Engdahl, 2015). When the magnitude uncertainty is above 0.7 m.u., the earthquake is listed in the supplementary catalogue. We agree that this approach, generally speaking, leads to higher uncertainties for pre-1976 earthquakes. However, either we ignore the magnitude uncertainty (as normally done in past catalogues) or we try to improve the parameters listed in the catalogue. Such an effort is summarized in Section 6 ("**Review of events that have already been part of the catalogue**"). There we outline the improvements achieved for events already listed in previous versions of the ISC-GEM Catalogue and show in Figure 22 how we have better constrained MS by adding a significant number of stations contributing toward MS re-computation. As the manuscript already contains many figures, we did not include the improvements in magnitude uncertainty for the revised earthquakes. This is shown in Fig. AR3. As we add station magnitudes we expect to better constrain our re-computed magnitudes and their uncertainties so that the Mw proxy uncertainties will also benefit. Section 8 ("**Outlook**") outlines our intention to continue looking for additional station data (see fourth bullet point), particularly for pre-1964 earthquakes, in order to better improve the magnitude parameters when necessary and possible.*

*Nevertheless, we thank the reviewer for this comment and, to remind the catalogue users of the limitations related to magnitude uncertainty, we have added a figure (Fig. 26 in the revised manuscript) to show the magnitude uncertainty in the ISC-GEM Catalogue. Main text and figure captions have been updated accordingly and we also added the following short paragraph at the end of Section 7: "In addition, users should be aware that the magnitude uncertainty for pre-digital earthquakes is inevitably larger than for earthquakes in the GCMT era (from 1976 onwards). The timeline of the Mw uncertainty in the ISC-GEM Main Catalogue is shown in Fig. 26. This is to further remind users of the full catalogue that, for patterns of seismicity studies, they should be aware of the larger magnitude uncertainty in the first part of last century."*

**AC: Both of these effects arise from enforcing a consistent treatment of event observations for the entire catalog duration, and it is not clear that there is any better way to handle the differences in data quality over time. However, these effects also make it rather difficult to use the full catalog, as they have the potential to hide real signals under selection limits and magnitude uncertainties. It might be useful for the commentary to point out that, in addition to the acknowledged likelihood of missing events early in the century, these missing events alias into an unlikely-to-be-real trend in number of large events and therefore seismic productivity. Alternatively, the commentary could include a short additional section that specifically outlines the implications of the event counts (as, for example, clearly shown in figure 1) to hypotheses about temporal patterns.**

*AR: Section 7 ("**Overall status of the new (V5) ISC-GEM Catalogue**") is the place where such concerns are pointed out and discussed. Figure 24 is very rich in information and it has been produced with the intention of also showing the limitations of the catalogue. In addition, as stated in the manuscript, it is clear that the period before 1964 is not as complete as from 1964 onwards (as one can infer by looking at Figure 24). Besides, the discussion in terms of Mc fluctuations over time is already present in the main text, we have added, taking the RC into consideration, the following sentence at the end of the first bullet point of Section 7:*
*"The fluctuations over time of the number of earthquakes (i.e., variations of Mc) in the full catalogue (especially at the lower magnitudes, below 6.5) should be checked before using it in its current status for studies concerning temporal and seismicity patterns."*

**RC: Some note of how the ISC-GEM compares to other commonly used catalogs should be included in the same section. Such a commentary would help ensure that researchers using the catalog are sufficiently aware of its inevitable biases.**

*AR: such comparisons are already discussed in Di Giacomo et al. (2015a) and we feel that it would largely repeat previous findings. It may not be intentional but from the RC we also feel that the reviewer considers previous catalogues (e.g., PAGER-CAT) immune from any bias, but we believe we brought enough evidence (in previous papers as well as in this reply) of how important it is to re-assess the basic parameters characterizing an earthquake (i.e., location and magnitude), particularly before the 1960s. The need for improvement was recognized by the Scientific Board of the GEM Foundation, whose financial support for the production of the ISC-GEM Catalogue was later followed by USGS, NSF and other organizations.*

**RC: As long as studies using the ISC-GEM catalog either acknowledge the resulting systematic bias or use additional events from other catalogs to "fill-in", the impacts can be addressed, if not mitigated.**

*AR: Earthquakes not listed in the ISC-GEM Catalogue can be found in the comprehensive ISC Bulletin (www.isc.ac.uk) or other resources, such as PAGER-CAT. However, as explained earlier, there is a reason if a global earthquake (e.g., magnitude 5.5 and above) is not listed in the ISC-GEM Catalogue. The case of the 1908-12-12 12:08 earthquake is emblematic. We, therefore, suggest caution in adding events from other catalogues to address or mitigate the ISC-GEM Catalogue shortcomings, as such additions are not at all guaranteed to improve things and safeguard the ISC-GEM Catalogue users from obtaining biased or spurious results.*

**RC: However, any studies using the whole catalog without consideration of the selection bias may produce spurious results**

*AR: We believe that we have now sufficiently emphasized the limitations of the catalogue with the changes made to the text and the additional plot of the magnitude uncertainty.*

[Figure]

*Figure AR1: Magnitude comparisons between re-computed MS(ISC-GEM) and other catalogues: a) ABE; b) G&R; c) B&D; d) ROTHE*

[Figure]

*Figure AR2: Magnitude differences vs time between re-computed MS(ISC-GEM) and other cata-logues: a) ABE; b) G&R; c) B&D; d) ROTHE. For each subplot the histograms distribution of the magnitude differences are also shown along with the average and standard deviation.*

[Figure]

*Figure AR3: Uncertainty distribution of the ISC-GEM re-computed MS for the revised earthquakes shown in Fig. 22 of the manuscript (Section 6). The grey and black histograms are for the earth-quakes before and after revision, respectively. Note the overall reduction in MS uncertainty after the revision (black histogram), which, in turn, translates in a lower Mw proxy uncertainty.*

[revised manuscript text omitted]

We  based the selection of the extension events on combinations of the number of body-wave arrival times and the number of stations  supplying amplitude data.  Considering that our relocation approach (Bondár et al., 2015) relies largely on teleseismic observations  above 18° distance) and  the magnitude re-assessment (Di Giacomo et al., 2015a) on the availability of three (or two in some case) station magnitudes, we first excluded events having no teleseismic phases and less than two  stations contributing with amplitudes. After this first cut, we further excluded earthquakes having a limited number of body-wave arrival times  and less than two to three stations with amplitudes. These are earthquakes for which we could not obtain a reliable solution (due to poor station coverage and/or arrival times) after preliminary relocation attempts. It is worth pointing out we have tried to be as comprehensive and conservative as possible by not rejecting all poorly constrained relocations (see next section). Also, we included all extension events between 1953 and 1956  available in the ISS (due to their small number. , see Fig. 1) and well recorded earthquakes but without amplitudes. As a result, out of the 19,341 extension events between 1920 and 1959 we relocated 11,572. The annual numbers are shown in Fig. 6, where the variations are linked to the state of the global network during those years and the operational practice changes at the ISS, as mentioned earlier.

**3.2 Relocations**

The location reassessment of previous hypocentres (from ISS or other authors adopted by it) of the selected extension events  is one of the fundamental tasks of this work. The relocations are obtained by closely following the approach described by Bondár et al. (2015). 
[revised manuscript text omitted]
 Considering all sources depicted in Fig. 13, Fig. 14 shows the overall annual counts of for the number of stations, phases and, finally, the box-and-whisker plot of the annual number of stations per event in each year. A significant dip is present in the station data between 1908 and 1912 since the station (and location) sources available to us for these years are not as comprehensive as ISA or BAAS/ISS. The box-and-whisker plot of Fig. 14 also shows that several earthquakes have none to three stations associated (59 from the augmented Centennial Catalogue, BAAS and ISS and 116 from the newly added ones). Obviously, the limitations in the collection of station data influenced the earthquakes that we finally

selected for processing and the quality of the relocations/magnitude re-assessment. The results are discussed in the next two subsections.

**4.2 Relocations**

Not all extension earthquakes have sufficient station data to perform a relocation using our approach. First, we have discarded 175 earthquakes having less than four stations, as pointed out earlier. We then progressively discarded another 650 as either the relocation failed or was considered unreliable. We may to go back to the discarded earthquakes if additional station data becomes available to us. In the end, we accepted the relocation for 1,110 out of the 1,935 extension earthquakes. Fig. 15 shows the annual counts of the relocated extension earthquakes 1904-1919. Note the dip in the annual number of the relocated extension earthquakes for 1908-1912, reflecting the absence (to our knowledge) of a comprehensive global bulletin between ISA and BAAS.

As in Fig. 7, Fig. 16 shows the box-and-whisker plots of NDEFSTA and SGAP. For this period the relocations are usually based on a small number of stations (median between 6 and 16) resulting in a large SGAP (median between 201° and 310º), even during the years covered by BAAS and ISS. Fig. 17 shows the median location, depth and origin time differences between previous (see Fig. 12) and ISC-GEM locations. The median location differences oscillate between 70 and 205 km, with large differences above one thousand km for 46 earthquakes (16 above 2000 and 4 above 3000 km). Such large location differences can occur for various reasons (from typos in the latitude/longitude of previous locations to poorly recorded earthquakes having low confidence locations). One extreme example is the epicentre change from Bristol Bay, off-shore Alaska, (G&R location) to off-shore Jamaica (ISC-GEM location) for an event that occurred the $22^{nd}$ August 1907 ($\sim$$22^h23^m$). The reason for such a large difference originates from the fact that G&R ignored the report that the event was felt in Kingston (see, e.g., ISA, 1907, part B, p. 73) and preferred to fit the phase data to an intermediate-depth event off-shore Alaska. As for 1920-1959, most of the earthquakes have no depth resolution and the previous depths were largely unknown or set to zero and this occasionally results in large depth changes (±100 and ±300 km for 51 and 10 earthquakes, respectively). Fig. 17 also shows the box-and-whisker plot of the origin time (OT) differences in each year. We show the OT differences because in this period (particularly before BAAS) the OT listed in the previous location sources was at time truncated to the minute or with some minute error that we were able to address thanks to the stations data we digitized. Although $\sim$90% of the OT differences are within 1 minute, some large OT changes of ±5 minutes or more occur for 16 earthquakes (eight originally from ABE).

Similar to the 1920-1959 period, we assigned location quality flag D if the location was not constrained well enough. This time this task was done not only by considering the usual criteria (see 3.2) but also consulting available information on the earthquake's effects (e.g., tsunami, damage). In this respect we made systematic use of the earthquake effect information available in UTSU and NGCDC. Table 4 summarizes the location and depth quality flags for the relocated extension events between 1904 and 1919. The limitations of the global network in this period are generally more prominent than for 1920-1959 and this translates in most of the earthquakes having location and depth quality C and about 246 of them have location quality D. As for the discarded earthquakes, if additional station data becomes available we will try to improve the location quality and eventually move some of the location flag D earthquakes from the Supplementary to the Main catalogue. As for Fig. 9,

Fig. 18 compares the previous (before) and ISC-GEM locations (after) on global maps where, again, a general improvement in the earthquakes distribution along plate boundaries is delineated. This is particularly the case for several global earthquakes along the subduction zone of the Pacific and Indian oceans whose previous locations were hundreds of km away from plate boundaries.

**4.3   Magnitude re-assessment**

Even for this period the magnitude re-assessment is mostly based on our re-computed MS. Following the same procedures described earlier, we obtained 927 MS for the relocated extension earthquakes, as shown in Fig. 19. For 500 of them we have computed a magnitude for the first time (in our record). Notably, for 137 earthquakes MS < 5.5, whereas MS $\geq$ 6.5 for 306 of them and > 7.5 for 12 of them. The latter includes six earthquakes originally from GUTE, four from ABE and two from BAAS that were not selected for V1 because the magnitudes available were not considered reliable or were below 7.5 (the original cut-off magnitude for the V1 selection before ISS started in 1918). Nearly all earthquakes with MS < 5.5 occurred in the Euro-Mediterranean area (because in this period the stations contributing with surface wave amplitudes are strongly concentrated in Europe, see Fig. 13).  In 1904  the collection of surface wave amplitudes is limited to two stations, GTT (Göttingen) and POT (Potsdam), until December, when we could also add data from station LEI (Leipzig). Consequently, in 1904 
[revised manuscript text omitted]
 fluctuations over time of the number of earthquakes (i.e., variations of Mc) in the full catalogue (especially at the lower magnitudes, below ∼6.5) should be checked before using it in its current status for studies concerning temporal and seismicity patterns;

- the number of intermediate-depth (between 60 and 300 km) and deep (≥ 300 km) earthquakes per year before the 1950s-1960s is significantly smaller compared to more recent decades. The reason is not fully clear and will be a matter for further investigation (see 8). Most likely, it is the result of a combination of factors, which include the detection capability for moderate deep-focus earthquakes of analog seismographs (see, e.g., Kanamori, 1988) deployed around the world before the 1950s, the lack of stations close to subduction zones for many decades (Figs. 2 and 13), and the earthquake selection criteria. For global earthquakes, instruments such as the Wiechert, Bosh-Omori, Maika and Galitzin were able to record surface wave signals (medium period range, centred around 20 s) better than body-waves (higher frequency signals, particularly P-waves, from around 10 s and below). The effect could have been that many stations would not report station data for moderate deep-focus earthquakes and, therefore, the ISS would not compile data for such earthquakes (i.e., the earthquake would not be recorded). The selection criteria could also play a role although the earthquakes not selected for processing either lack station data (and depth resolution) or, more importantly, are usually too small to account for the small number of deep-focus earthquakes depicted in Fig. 25.

In addition, users should be aware that the magnitude uncertainty for pre-digital earthquakes is inevitably larger than for earthquakes in the GCMT era (from 1976 onwards). The timeline of the Mw uncertainty in the ISC-GEM Main Catalogue is

shown in Fig.26. This is to further remind users of the full catalogue that, for patterns of seismicity studies, they should be aware of the larger magnitude uncertainty in the first part of last century.

[revised manuscript text omitted]

**Table 1.** Criteria to assign the location quality flags for location, depth and magnitude. SGAP is the secondary azimuthal gap, GTCAND denotes a high confidence location accuracy that makes the event a candidate for the IASPEI  Reference Event List (Bondár and McLaughlin, 2009, see also www.isc.ac.uk/gtevents), depdp is the depth constrained by depth phases (if available), NSTA10 is the number of stations within 10 km and NSTA(local) is the number of stations within 150 km (Bondár and Storchak, 2011). MS is considered well constrained when it is obtained from more than 4 stations, within 5.5-7.5, and has uncertainty $\leq 0.2$. See text for details.

| Quality flag | Location | Depth | Magnitude (Mw, direct or proxy) |
|---|---|---|---|
| A | SGAP < 120 && eccentricity < 0.75 or GTCAND | depdp or GTCAND or NSTA10 | GCMT |
| B | SGAP < 160 | NSTAlocal > 2 | Literature or Proxy based on well constrained MS |
| C | Other cases | Other cases | Literature or Proxy based on poorly constrained MS or Proxy based on mb |
| D | Manually assigned | Manually assigned | No magnitude or Proxy uncertainty > 0.7 or Mw proxy based on MS < 5 |

**Table 2.** Summary of the location and depth quality flags for the extension events between 1920 and 1959.

| Quality flag | Count for location | Count for depth |
|---|---|---|
| A | 1744 | 1886 |
| B | 3315 | 479 |
| C | 6431 | 9208 |
| D | 83 | 0 |

t

**Table 3.** Summary of the magnitude quality flags for the relocated extension events between 1920 and 1959. Included in the D flag are 4984 events where no magnitude was recomputed.

| Quality flag | Count for magnitude |
|---|---|
| A | 0 |
| B | 3030 |
| C | 2824 |
| D | 5719 |

t

**Table 4.** Summary of the location and depth quality flags for the relocated extension events between 1904 and 1919.

| Quality flag | Count for location | Count for depth |
|---|---|---|
| A | 14 | 6 |
| B | 75 | 2 |
| C | 775 | 1102 |
| D | 246 | 0 |

t

**Table 5.** As for Table 3 but for 1904-1919. Included in the D flag are 183 events where no magnitude was recomputed.

| Quality flag | Count for magnitude |
|---|---|
| A | 0 |
| B | 420 |
| C | 427 |
| D | 262 |

t

**Table 6.** Number of earthquakes added between 2010 and 2014.

| Year | Count |
|---|---|
| 2010 | 504 |
| 2011 | 672 |
| 2012 | 414 |
| 2013 | 484 |
| 2014 | 521 |

---

## Author Comment (AC2) · 26 Sep 2018

**Author's Response to the Reviewer's Comments #2 (PAPER: The ISC-GEM Earthquake Catalogue (1904-2014): status after the Extension Project, https://doi.org/10.5194/essd-2018-59)**

Domenico Di Giacomo        E. Robert Engdahl        Dmitry A. Storchak

September 26, 2018

**Response to general comments**

We thank the reviewer for pointing out some presentation issue and sentences that need to be clarified. Below we reply in detail to each point showing the Reviewer Comment (RC) and the Author Response (AR) in italic.

**RC: Most of my comments, which are detailed below, concern presentation issues. In particular, I find that the reference throughout the article to the 4-year project and its stages is somewhat cumbersome and uninteresting for the reader. While such references would be adequate in a technical project report, readers of the article will be interested in the catalog and not on how the project was timed and developed. I thus encourage the authors to eliminate from the article references to the yearly development of the extension project, and rather focus only on the catalogue analysis.**

*AR: We understand the point of the reviewer, and share in part his reasoning. However, there are a few elements that motivated us to structure the manuscript around the stages of the Extension Project: 1) there is no technical report that would allow readers to appreciate the time and effort put into the Extension of the ISC-GEM Catalogue; 2) researchers often overlook the amount of time an organization invests in terms of manual and repetitive work necessary to produce a dataset that the community can use for various types of research for many years to come. If not detailed here, the project stages and the effort put into it will not be documented, and that may be forgotten in the near future. Therefore we keep the references to the yearly developments. However we take on board some of the reviewer suggestions related to this point, as detailed later in our replies to his minor comments.*

**RC: I also find a bit difficult to follow the description of the data and workflow. It would help if the authors could add a flowchart with the processing. The dataset could be identified at each step as extension-0, extension-1, extension-x, extension-final, and the processing steps could be identified as well. A table could accompany the flowchart, including the number of events after each processing step. If needed, adapt for the different time periods.**

*AR: This general comment is related to a later minor RC ("**P6, L7-12. The criteria are somewhat vague. Is it possible to detail, for example in a table?**"). We believe that the workflow is already explained in Section 2 and the structure of Section 3 and 4 follows such workflow (i.e. data collection, relocation, magnitude re-assessment). We, therefore, consider the addition of a flowchart to be unnecessary. We agree though that the text describing the earthquake selection criteria based on the collected data needs to be clarified. For the sake of the reader, we have simplified it, but without adding a table as suggested (it would be too convoluted with the risk of making the situation even worse). The updated text at P6, L6-16 of the original manuscript now reads:*

*"We based the selection of the extension events on combinations of the number of body-wave arrival times and the number of stations supplying amplitude data. Considering that our relocation approach (Bondár et al., 2015) relies largely on teleseismic observations (i.e., above 18° distance), and the magnitude re-assessment (Di Giacomo et l., 2015a) on the availability of three (or two in some case) station magnitudes, we first excluded events having no teleseismic phases and less than two stations contributing with amplitudes. After this first cut, we further excluded earthquakes having a limited number of body-wave arrival times and less than two to three stations with amplitudes. These are earthquakes for which we could not obtain a reliable solution (due to poor station coverage and/or arrival times) after preliminary relocation attempts. It is worth pointing out we have tried to be as comprehensive and conservative as possible by not rejecting all poorly constrained relocations (see next section). Also, we included all extension events between 1953 and 1956 available in the ISS (due to their small number, see Fig. 1) and well recorded earthquakes but without amplitudes. As a result, out of the 19,341 extension events between 1920 and 1959 we relocated 11,572. The annual numbers are shown in Fig. 6, where the variations are linked to the state of the global network during those years and the operational practice changes at the ISS, as mentioned earlier."*

**RC: The article is written in easily understandable English, but still some sentences require re-reading. If possible, I would recommend that the article be carefully edited for English.**

*AR: Both the sentences mentioned in the minor comments and the main text has been proof-read by a native English speaker in order to streamline the text. Changes are reproduced below and highlighted in the revised version of the manuscript.*

**Response to minor comments**

**RC: Page 1, Lines 1-5: Too long sentence. Becomes confusing. Rephrase/clarify.**

*AR: we have update the sentence as "We outline the work done to extend and improve the ISC-GEM Global Instrumental Earthquake Catalogue, a dataset which was first released in 2013 (Storchak et al., 2013, 2015). In its first version (V1) the catalogue included global earthquakes selected according to time dependent cut-off magnitudes: 7.5 and above between 1900 and 1918 (plus significant 6.5 and above continental earthquakes); 6.25 between 1918 and 1959; 5.5 between 1960*

*and 2009. Such selection criteria were dictated by time and resource limitations.”*

**RC: P1, L 11-13: Include a brief explanation as to why the ISC-GEM catalog now starts only in 1904 rather than 1900 (as originally).**

*AR: we have updated the sentence as “Compared to the 2013 release, we eliminated earthquakes during the first 4 years (1900-1903) of the catalogue (lack of reliable station data), added approximately 12,000 and 2,500 earthquakes before 1960 and between 2010 and 2014, respectively, and improved the solution for approximately 2,000 earthquakes already listed in previous versions.”*

**RC: P2, L7: “large earthquakes pre-1964 and from 1964 onwards,”- confusing, rephrase.**

*AR: we have updated the sentence as “For example, the Centennial Catalogue lists both locations from various catalogues (including the ones mentioned above) and re-computed ones (from 1964 onwards and only for selected large earthquakes between 1918 and 1964) using the Engdahl et al. (1998) methodology (normally referred to as EHB)”.*

**RC: P3, L24. I would suggest to remove the text and table (Table 1) about the timing of the project, unless there is a good reason to show it.**

*AR: We removed Table 1 from the manuscript and included such information in Figure 1, as also suggested by the reviewer in a later point. Caption and text updated accordingly.*

**RC: P5, L3: Remove the reference to the phase of the project, leave only the reference to catalogue time period (1920-1959).**

*AR: Done. For consistency, we have also updated the title of Section 4.*

**RC: P5, L12-14. Confusing. Rephrase/clarify.**

*AR: we have updated the sentence as “For example, between 1950 and 1952 the annual number of extension events in the ISS is between 782 and 1384, and such numbers are above the annual number of earthquakes of magnitude 5.5 and above in the ISC-GEM Catalogue in recent years (e.g., the largest annual number of earthquakes is 654 for 2011). This means that a subset of the extension events in 1950-1952 should not be part of the ISC-GEM Catalogue as it falls below the cut-off magnitude of 5.5.”*

**RC: P6, L7-12. The criteria are somewhat vague. Is it possible to detail, for example in a table?**

*AR: We replied this RC in the Response to General Comments.*

**RC: P6, L18-19. Confusing. Rephrase/clarify.**

*AR: we have updated the sentence as "The location reassessment of previous hypocenters (ISS or other authors adopted by it) of the selected extension events is one of the fundamental tasks of this work. The relocations are obtained by closely following the approach described by Bondár et al. (2015)."*

**RC: P7, L5. Replace "recurrent" by "frequent".**

*AR: Done.*

**RC: P9, L16, and later. I find it confusing to use the word "counts" to refer to the number of locations available in each source. Maybe "reported locations" or some different word/expression is more adequate. "Counts" seems too generic.**

*AR: We believe the RC refers to L11 instead of L16. We normally use "count" in the sense of sum/total, and we believe it is appropriate in our context (annual number). However, we have rephrased slightly the sentence as "Considering all sources depicted in Fig. 14, Fig. 13 shows the overall annual counts for the number of stations, phases and, finally, the box-and-whisker plot of the annual number of stations per event.*

**RC: P11, L28-29. Rephrase/clarify.**

*AR: we have updated the sentence as "In 1904 the collection of surface wave amplitudes is limited to two stations, GTT (Göttingen) and POT (Potsdam), until December, when we could also add data from station LEI (Leipzig). Consequently, in 1904 we were able to re-compute MS for three earthquakes only, all occurring in December."*

**RC: P14, L22. Careful with referring to "historical" earthquakes, as this expression is typically used for earthquakes documented historically but not recorded instrumentally. Maybe better to use "pre-digital", if this is the case?**

*AR: Agreed. We replaced "historical" with "pre-digital" throughout the text.*

**RC: Figure 1. Identify in the figure the different time periods referred to in the text (P3,L14).**

*AR: Done. See new Figure 1 in the revised manuscript and updated caption.*

**RC: Figure 6. It would be nice to plot, behind this histogram, a second histogram showing the initial number of extension events in the period studied.**

*AR: Figure 1 already shows the initial number of extensions events.*

**RC: Figure 23. It would be interesting to add to this figure a similar one, but showing only the earthquakes with well-constrained locations.**

*AR: The figure includes only 240 earthquakes from the supplementary catalogue (i.e., with not-so-well-constrained location), thus adding a figure as suggested would not add anything significant to the manuscript as the figures look basically the same. Considering the large number of figure already present in the manuscript we prefer not to add the figure suggested.*

**RC: Acknowledgements: Josep Batlló, there is no "h" at the end of "Josep", and there are two "l" and one "t" in "Batlló" (!).**

*AR: Our apologies for the typos, name corrected.*

[revised manuscript text omitted]
  pre-digital period between 1920 and 1959. Note that throughout this work we consider as pre-digital earthquakes

10 those that occurred before 1964 (i.e., before the beginning of the ISC Bulletin).

**3.1   Station data collection and earthquake selection**

The variations in the annual number of the extension events shown in Fig. 1 are the result of various factors. For example, a significant increase in the annual number of events can be seen in 1918 coinciding with the beginning of the ISS, whereas a dip in the late 1930s to mid-1940s is associated with the disruption caused by World War II (more details later) and another dip

15 in the mid-1950s is due to the censoring introduced by ISS procedures (more details at page 3 of the ISS, 1953) to reduce the workload. The annual variations in the number of the extension events also introduces an issue in selecting earthquakes for the ISC-GEM Catalogue. For example, between 1950 and 1952  the annual number of extension events in the ISS  is between 782 and  1384, and such numbers are above the annual number of earthquakes of magnitude 5.5 and above in the ISC-GEM Catalogue in recent years  (e.g., the largest annual number of

20 earthquakes is 654 for 2011). This means that a  subset of the extension events in 1950-1952 should not be part of the ISC-GEM Catalogue as  it falls below the cut-off magnitude of 5.5. However, since, as mentioned earlier, about 60% of such events have no magnitude information in our database, we could not use the original magnitude criteria of 5.5. Thus, we decided to base our selection criteria using for each extension event both the distribution of stations in the ISS and the number of stations contributing with amplitudes/periods for magnitude re-computation. This required a major effort to digitize both all

25 ISS pages (not available in any electronic format) and amplitude and period pairs (of surface waves, in particular) from the station/network printed bulletins (Di Giacomo et al., 2015b) for all extension events. Here we briefly summarize the station data collected for the extension events and highlight some features that are relevant to the ISC-GEM Catalogue.

Fig. 2 shows the distribution of stations listed in the ISS for each decade (1920s, 1930s, 1940s, 1950s) color-coded by their body-wave arrivals contribution to the extension events along with the annual number of stations and body-wave arrivals

30 digitized from the ISS. The number of stations listed in the ISS generally increased from the 1920s to the late 1930s before World War II affected various seismic stations and it is only around 1953 that the station contribution improved significantly. The box-and-whisker plot of Fig. 3 summarizes the median number of stations per event in each year. It shows that only a

limited number of stations (median number ranging from 9 to 26) are usually associated to the extension events until 1952, whereas from 1953 onwards there is a general improvement in this respect (median number of stations ranging from 66 to 99). Another relevant feature to point out is the uneven station distribution, with Europe showing the highest density particularly before the 1950s, and the lack of stations in Africa and vast parts of the southern hemisphere.

Fig. 4 and Fig. 5, similarly to Fig. 2 and Fig. 3, show the distribution of stations contributing with amplitudes for each decade and the median number of stations supplying amplitudes in each year, respectively. The number of stations reporting amplitudes increased until World War II, dropped in the 1940s, and improved significantly from 1953. European and many Russian stations are the most important contributors to amplitude readings compared to stations in other continents, except for La Paz (LPZ, Observatorio San Calixto, Bolivia) and Riverview College (RIV, Sydney, Australia) from the Jesuit seismic network (Udías and Stauder, 1996). The number of stations per event contributing with amplitudes ranges from 0 to above 40, with the median per year oscillating from 0 to 6 (Fig. 5).

We based the selection of the extension events on combinations of the number of body-wave arrival times and the number of stations supplying amplitude data. Considering that our relocation approach (Bondár et al., 2015) relies largely on teleseismic observations above 18° distance) and the magnitude re-assessment (Di Giacomo et al., 2015a) on the availability of three (or two in some case) station magnitudes, we first excluded events having no teleseismic phases and less than two stations contributing with amplitudes. After this first cut, we further excluded earthquakes having a limited number of body-wave arrival times and less than two to three stations with amplitudes. These are earthquakes for which we could not obtain a reliable solution (due to poor station coverage and/or arrival times) after preliminary relocation attempts. It is worth pointing out we have tried to be as comprehensive and conservative as possible by not rejecting all poorly constrained relocations (see next section). Also, we included all extension events between 1953 and 1956 available in the ISS (due to their small number, see Fig. 1) and well recorded earthquakes but without amplitudes. As a result, out of the 19,341 extension events between 1920 and 1959 we relocated 11,572. The annual numbers are shown in Fig. 6, where the variations are linked to the state of the global network during those years and the operational practice changes at the ISS, as mentioned earlier.

**3.2 Relocations**

The location reassessment of previous hypocentres (from ISS or other authors adopted by it) of the selected extension events is one of the fundamental tasks of this work. The relocations are obtained by closely following the approach described by Bondár et al. (2015). 
[revised manuscript text omitted]
 Considering all sources depicted in Fig. 13, Fig. 14 shows the overall annual counts of for the number of stations, phases and, finally, the box-and-whisker plot of the annual number of stations per event in each year. A significant dip is present in the station data between 1908 and 1912 since the station (and location) sources available to us for these years are not as comprehensive as ISA or BAAS/ISS. The box-and-whisker plot of Fig. 14 also shows that several earthquakes have none to three stations associated (59 from the augmented Centennial Catalogue, BAAS and ISS and 116 from the newly added ones). Obviously, the limitations in the collection of station data influenced the earthquakes that we finally

selected for processing and the quality of the relocations/magnitude re-assessment. The results are discussed in the next two subsections.

**4.2 Relocations**

Not all extension earthquakes have sufficient station data to perform a relocation using our approach. First, we have discarded 175 earthquakes having less than four stations, as pointed out earlier. We then progressively discarded another 650 as either the relocation failed or was considered unreliable. We may to go back to the discarded earthquakes if additional station data becomes available to us. In the end, we accepted the relocation for 1,110 out of the 1,935 extension earthquakes. Fig. 15 shows the annual counts of the relocated extension earthquakes 1904-1919. Note the dip in the annual number of the relocated extension earthquakes for 1908-1912, reflecting the absence (to our knowledge) of a comprehensive global bulletin between ISA and BAAS.

As in Fig. 7, Fig. 16 shows the box-and-whisker plots of NDEFSTA and SGAP. For this period the relocations are usually based on a small number of stations (median between 6 and 16) resulting in a large SGAP (median between 201° and 310º), even during the years covered by BAAS and ISS. Fig. 17 shows the median location, depth and origin time differences between previous (see Fig. 12) and ISC-GEM locations. The median location differences oscillate between 70 and 205 km, with large differences above one thousand km for 46 earthquakes (16 above 2000 and 4 above 3000 km). Such large location differences can occur for various reasons (from typos in the latitude/longitude of previous locations to poorly recorded earthquakes having low confidence locations). One extreme example is the epicentre change from Bristol Bay, off-shore Alaska, (G&R location) to off-shore Jamaica (ISC-GEM location) for an event that occurred the 22$^{nd}$ August 1907 (~22$^h$23$^m$). The reason for such a large difference originates from the fact that G&R ignored the report that the event was felt in Kingston (see, e.g., ISA, 1907, part B, p. 73) and preferred to fit the phase data to an intermediate-depth event off-shore Alaska. As for 1920-1959, most of the earthquakes have no depth resolution and the previous depths were largely unknown or set to zero and this occasionally results in large depth changes (±100 and ±300 km for 51 and 10 earthquakes, respectively). Fig. 17 also shows the box-and-whisker plot of the origin time (OT) differences in each year. We show the OT differences because in this period (particularly before BAAS) the OT listed in the previous location sources was at time truncated to the minute or with some minute error that we were able to address thanks to the stations data we digitized. Although ~90% of the OT differences are within 1 minute, some large OT changes of ±5 minutes or more occur for 16 earthquakes (eight originally from ABE).

Similar to the 1920-1959 period, we assigned location quality flag D if the location was not constrained well enough. This time this task was done not only by considering the usual criteria (see 3.2) but also consulting available information on the earthquake's effects (e.g., tsunami, damage). In this respect we made systematic use of the earthquake effect information available in UTSU and NGCDC. Table 4 summarizes the location and depth quality flags for the relocated extension events between 1904 and 1919. The limitations of the global network in this period are generally more prominent than for 1920-1959 and this translates in most of the earthquakes having location and depth quality C and about 246 of them have location quality D. As for the discarded earthquakes, if additional station data becomes available we will try to improve the location quality and eventually move some of the location flag D earthquakes from the Supplementary to the Main catalogue. As for Fig. 9,

Fig. 18 compares the previous (before) and ISC-GEM locations (after) on global maps where, again, a general improvement in the earthquakes distribution along plate boundaries is delineated. This is particularly the case for several global earthquakes along the subduction zone of the Pacific and Indian oceans whose previous locations were hundreds of km away from plate boundaries.

**4.3 Magnitude re-assessment**

Even for this period the magnitude re-assessment is mostly based on our re-computed MS. Following the same procedures described earlier, we obtained 927 MS for the relocated extension earthquakes, as shown in Fig. 19. For 500 of them we have computed a magnitude for the first time (in our record). Notably, for 137 earthquakes MS < 5.5, whereas MS $\geq$ 6.5 for 306 of them and > 7.5 for 12 of them. The latter includes six earthquakes originally from GUTE, four from ABE and two from BAAS that were not selected for V1 because the magnitudes available were not considered reliable or were below 7.5 (the original cut-off magnitude for the V1 selection before ISS started in 1918). Nearly all earthquakes with MS < 5.5 occurred in the Euro-Mediterranean area (because in this period the stations contributing with surface wave amplitudes are strongly concentrated in Europe, see Fig. 13).  In 1904 the collection of surface wave amplitudes is limited to two stations, GTT (Göttingen) and POT (Potsdam), until December, when we could also add data from station LEI (Leipzig). Consequently, in 1904 
[revised manuscript text omitted]
 fluctuations over time of the number of earthquakes (i.e., variations of Mc) in the full catalogue (especially at the lower magnitudes, below ∼6.5) should be checked before using it in its current status for studies concerning temporal and seismicity patterns;

- the number of intermediate-depth (between 60 and 300 km) and deep ($\geq$ 300 km) earthquakes per year before the 1950s-1960s is significantly smaller compared to more recent decades. The reason is not fully clear and will be a matter for further investigation (see 8). Most likely, it is the result of a combination of factors, which include the detection capability for moderate deep-focus earthquakes of analog seismographs (see, e.g., Kanamori, 1988) deployed around the world before the 1950s, the lack of stations close to subduction zones for many decades (Figs. 2 and 13), and the earthquake selection criteria. For global earthquakes, instruments such as the Wiechert, Bosh-Omori, Maika and Galitzin were able to record surface wave signals (medium period range, centred around 20 s) better than body-waves (higher frequency signals, particularly P-waves, from around 10 s and below). The effect could have been that many stations would not report station data for moderate deep-focus earthquakes and, therefore, the ISS would not compile data for such earthquakes (i.e., the earthquake would not be recorded). The selection criteria could also play a role although the earthquakes not selected for processing either lack station data (and depth resolution) or, more importantly, are usually too small to account for the small number of deep-focus earthquakes depicted in Fig. 25.

In addition, users should be aware that the magnitude uncertainty for pre-digital earthquakes is inevitably larger than for earthquakes in the GCMT era (from 1976 onwards). The timeline of the Mw uncertainty in the ISC-GEM Main Catalogue is

shown in Fig.26. This is to further remind users of the full catalogue that, for patterns of seismicity studies, they should be aware of the larger magnitude uncertainty in the first part of last century.

[revised manuscript text omitted]

**Table 1.** Criteria to assign the location quality flags for location, depth and magnitude. SGAP is the secondary azimuthal gap, GTCAND denotes a high confidence location accuracy that makes the event a candidate for the IASPEI  Reference Event List (Bondár and McLaughlin, 2009, see also www.isc.ac.uk/gtevents), depdp is the depth constrained by depth phases (if available), NSTA10 is the number of stations within 10 km and NSTA(local) is the number of stations within 150 km (Bondár and Storchak, 2011). MS is considered well constrained when it is obtained from more than 4 stations, within 5.5-7.5, and has uncertainty $\leq 0.2$. See text for details.

| Quality flag | Location | Depth | Magnitude (Mw, direct or proxy) |
|---|---|---|---|
| A | SGAP < 120 && eccentricity < 0.75 or GTCAND | depdp or GTCAND or NSTA10 | GCMT |
| B | SGAP < 160 | NSTAlocal > 2 | Literature or Proxy based on well constrained MS |
| C | Other cases | Other cases | Literature or Proxy based on poorly constrained MS or Proxy based on mb |
| D | Manually assigned | Manually assigned | No magnitude or Proxy uncertainty > 0.7 or Mw proxy based on MS < 5 |

**Table 2.** Summary of the location and depth quality flags for the extension events between 1920 and 1959.

| Quality flag | Count for location | Count for depth |
|---|---|---|
| A | 1744 | 1886 |
| B | 3315 | 479 |
| C | 6431 | 9208 |
| D | 83 | 0 |

t

**Table 3.** Summary of the magnitude quality flags for the relocated extension events between 1920 and 1959. Included in the D flag are 4984 events where no magnitude was recomputed.

| Quality flag | Count for magnitude |
|:---:|:---:|
| A | 0 |
| B | 3030 |
| C | 2824 |
| D | 5719 |

t

**Table 4.** Summary of the location and depth quality flags for the relocated extension events between 1904 and 1919.

| Quality flag | Count for location | Count for depth |
|:---:|:---:|:---:|
| A | 14 | 6 |
| B | 75 | 2 |
| C | 775 | 1102 |
| D | 246 | 0 |

t

**Table 5.** As for Table 3 but for 1904-1919. Included in the D flag are 183 events where no magnitude was recomputed.

| Quality flag | Count for magnitude |
|:---:|:---:|
| A | 0 |
| B | 420 |
| C | 427 |
| D | 262 |

t

**Table 6.** Number of earthquakes added between 2010 and 2014.

| Year | Count |
|:---:|:---:|
| 2010 | 504 |
| 2011 | 672 |
| 2012 | 414 |
| 2013 | 484 |
| 2014 | 521 |